# Enhancing chemotherapy response through augmented synthetic lethality by co-targeting nucleotide excision repair and cell-cycle checkpoints

Yi Wen Kong[1,2,15], Erik C. Dreaden[1,3,9,15], Sandra Morandell[1,10], Wen Zhou[1,4], Sanjeev S. Dhara [1], Ganapathy Sriram [1,2], Fred C. Lam [1,2,5], Jesse C. Patterson[1,2], Mohiuddin Quadir[1,3,11], Anh Dinh[1], Kevin E. Shopsowitz[1,3], Shohreh Varmeh[1,2], Ömer H. Yilmaz [1,6], Stephen J. Lippard[1,4], H. Christian Reinhardt[1,12,13,14], Michael T. Hemann[1,2], Paula T. Hammond [1,3 ✉] & Michael B. Yaffe [1,2,6,7,8 ✉]

In response to DNA damage, a synthetic lethal relationship exists between the cell cycle checkpoint kinase MK2 and the tumor suppressor p53. Here, we describe the concept of augmented synthetic lethality (ASL): depletion of a third gene product enhances a pre-existing synthetic lethal combination. We show that loss of the DNA repair protein XPA markedly augments the synthetic lethality between MK2 and p53, enhancing anti-tumor responses alone and in combination with cisplatin chemotherapy. Delivery of siRNA-peptide nanoplexes co-targeting MK2 and XPA to pre-existing p53-deficient tumors in a highly aggressive, immunocompetent mouse model of lung adenocarcinoma improves long-term survival and cisplatin response beyond those of the synthetic lethal p53 mutant/MK2 combination alone. These findings establish a mechanism for co-targeting DNA damage-induced cell cycle checkpoints in combination with repair of cisplatin-DNA lesions in vivo using RNAi nanocarriers, and motivate further exploration of ASL as a generalized strategy to improve cancer treatment.

[1] David H. Koch Institute for Integrative Cancer Research, Massachusetts Institute of Technology, Cambridge, MA 02139, USA. [2] Center for Precision Cancer Medicine, Massachusetts Institute of Technology, Cambridge, MA 02139, USA. [3] Department of Chemical Engineering, Massachusetts Institute of Technology, Cambridge, MA 02139, USA. [4] Department of Chemistry, Massachusetts Institute of Technology, Cambridge, MA 02139, USA. [5] Division of Neurosurgery, Hamilton General Hospital, McMaster University Faculty of Health Sciences, Hamilton, ON L8L 2X2, Canada. [6] Department of Biology, Massachusetts Institute of Technology, Cambridge, MA 02139, USA. [7] Department of Biological Engineering, Massachusetts Institute of Technology, Cambridge, MA 02139, USA. [8] Divisions of Surgical Oncology, Trauma, and Surgical Critical Care, Department of Surgery, Beth Israel Deaconess Medical Center, Harvard Medical School, Boston, MA 02215, USA. [9]Present address: Coulter Department of Biomedical Engineering, Georgia Institute of Technology and Emory University, Department of Pediatrics, Aflac Cancer and Blood Disorders Center, Children's Healthcare of Atlanta, Emory University School of Medicine, Atlanta, GA 30322, USA. [10]Present address: Molecular Health GmbH, 69115 Heidelberg, Germany. [11]Present address: Department of Coatings and Polymeric Materials, North Dakota State University, Fargo, USA. [12]Present address: Clinic I of Internal Medicine, University Hospital Cologne, Cologne, Germany. [13]Present address: Cologne Excellence Cluster in Cellular Stress Response in Aging-Associated Disorders (CECAD), University of Cologne, Cologne, Germany. [14]Present address: Center for Molecular Medicine Cologne, University of Cologne, Cologne, Germany. [15]These authors contributed equally: Yi Wen Kong, Erik C. Dreaden. ✉email: hammond@mit.edu; myaffe@mit.edu

Although there have been significant advances in the treatment of certain types of cancers with molecularly targeted therapies against driver oncogenes such as *EGFR*, *BRAF*, and *ALK*[1], for most tumor types, treatment with cytotoxic chemotherapeutic regimens, often containing platinum-based compounds, remains the frontline therapy[2]. Platinum drugs exert their effects by causing DNA damage, leading to the activation of a complex signaling network that induces cell cycle arrest and recruits DNA repair machinery to sites of damage[3–6]. If the damage exceeds the cells' ability for repair, cell death ensues, typically via apoptosis[5–8]. Thus, to enhance the effectiveness of DNA damaging chemotherapy for tumor cell killing, two general parallel approaches have been pursued: either disruption of cell cycle checkpoints by targeting critical effector kinases, or interfering with the process of DNA repair itself[5,6,9].

The concept of synthetic lethality (SL) holds great promise for the treatment of human cancers, best exemplified by the now widespread use of PARP inhibitors in BRCA mutant cancers. SL originally described a relationship between two genes, where alteration of either gene alone results in viable cells, but alteration (mutation, loss, or inhibition) of both genes simultaneously was lethal. The concept has now been extended to embrace synthetic lethal drug sensitivity, such as that observed with PARP inhibitors in combination with DNA-damaging chemotherapy in a variety of BRCA defective tumors[10]. Because BRCA mutations are observed in fewer than 10% of cancer patients (cBioPortal: 6.7%)[11–13] the identification of additional genes that share synthetic lethal sensitivity relationships with mutated oncogenes or tumor suppressors would greatly enhance the implementation of tumor cell-specific synthetic lethal sensitivity to improve an anticancer therapeutic response. We recently identified one such SL sensitivity relationship between loss or mutation of the tumor suppressor p53 and the cell cycle checkpoint effector kinase mitogen-activated protein kinase activated protein kinase-2 (MAPKAP kinase 2, MAPKAPK2, or MK2) in the context of DNA damaging chemotherapy[14–17]. Cancer cells that are defective in p53 function are deficient in their ability to transcriptionally upregulate the CDK inhibitor p21 after genotoxic stress. Therefore, compared to normal p53-proficient cells, p53-defective cells are more reliant on MK2 activity, which drives an alternative cell cycle checkpoint pathway that stabilizes the CKI inhibitors $p27^{Kip1}$ and Gadd45α in order to maintain $G_1/S$ and $G_2/M$ arrest after certain types of DNA damage[16,18]. Because most tumors are deficient in one or more aspects of the function of the p53 tumor suppressor, either as a consequence of mutations within p53, or impairment of upstream and downstream modulators of p53 activity[19], targeting MK2 has the potential to selectively enhance tumor cell killing without increasing the genotoxic effects of chemotherapy on normal p53-wild type tissues. Approximately, 50% of non-small cell lung cancers (NSCLC), for example, contain mutations, deletions, or truncations of p53 (cBioPortal: 55.5% of NSCLC and 45.7% of all tumor types)[11–13]. Genetic deletion of MK2 prior to tumor development dramatically enhances subsequent tumor killing by DNA-damaging chemotherapy selectively in tumor cells that lack functional p53, both in vitro and in vivo[14,17]. However, whether targeting of MK2 after the development of well-established tumors in vivo would also enhance the response to chemotherapy is unknown.

Platinum-containing compounds like cisplatin and carboplatin are amongst the most widely used cytotoxic chemotherapeutics for many tumor types, including ovarian cancer and lung adenocarcinoma. The major (>90%) adducts of cisplatin and carboplatin on DNA are 1,2-intrastrand d(GpG) and d(ApG) cross-links, which are repaired by the nucleotide excision repair (NER) pathway[5,6,20,21]. Tumor responses to cisplatin or carboplatin therefore depends on the levels of platinum–DNA adducts that are formed and the DNA repair capacity of the cell[20,21], as exemplified by the fact that testicular cancers, many of which have deficiencies in the NER pathway, respond extremely well to cisplatin with cure rates reaching ~95%[22–24]. However, in many other common tumor types such as NSCLC—the leading cause of cancer related death in the United States—the therapeutic efficacy of platinum-based DNA damaging agents is limited, with only about one third of patients receiving benefit[20,25]. It is well-established that tumor cells in culture can be sensitized to cisplatin through the inhibition of the NER pathway; however, utilizing this information for therapeutic gain in patients is limited by the fact that systemic inhibition of NER is highly toxic to host tissues[21,24].

Here, we describe the first example of an augmented synthetic lethal (ASL) sensitivity relationship between p53, MK2, and the DNA repair enzyme XPA. Simultaneous loss of MK2 and XPA increased tumor cell killing by cisplatin in p53-defective lung adenocarcinoma, the major subtype of NSCLC, cells relative to targeting either pathway alone due to hyperactivation of MK2 signaling in XPA-deficient cells. To leverage this ASL relationship for therapeutic gain, we utilized our recently developed RNAi delivery system to simultaneously co-target MK2 and XPA in p53-defective tumors in vivo, revealing superior tumor control with ASL targeting vs. SL targeting, and demonstrating a dramatic improvement in long-term survival. Together, these results highlight the utility of targeting an augmented SL relationship between cell cycle checkpoints and DNA repair to improve response to frontline chemotherapy in aggressive mouse models of human cancer.

## Results

**Cells defective in DNA repair by NER hyperactivate MK2 signaling.** Cisplatin-induced DNA damage primarily results from platinum (Pt)-mediated 1,2-intrastrand cross-links, and at a lower frequency, interstrand adducts. The former lesions are primarily repaired by the NER pathway, which includes both global genome repair (GG-NER) and transcription-coupled repair (TC-NER) (Fig. 1a)[24], and the latter by NER, translesion synthesis (TLS), homologous recombination (HR)[26], and the Fanconi anemia (FA) pathway[27]. Cells genetically lacking key NER proteins involved in GG-NER have reduced ability to repair certain types of Pt-induced lesions and related types of DNA cross-links[28], and have persistent DNA damage and error-prone repair, resulting in the human cancer-prone syndrome Xeroderma pigmentosum (XP)[3]. Defects in the CSB protein, which is involved in TC-NER, base-excision repair, and transcription, results in Cockayne syndrome (CS), characterized by mitochondrial dysfunction, premature aging, and neurodegeneration[29,30]. We hypothesized that defects in NER and the resulting persistent DNA damage accumulation, might further augment the activation of the MK2 signaling pathway in response to genotoxic stress. To investigate this, fibroblasts from XP or CS patients defective for individual proteins involved in GG-NER and/or TC-NER, and from a healthy control, were treated with increasing doses of cisplatin for 24 h, lysed, and probed for MK2 pathway activation by immunoblotting for the active, phosphorylated form of MK2[14]. As shown in Fig. 1b and Supplementary Fig. 1A, cells deficient in the NER proteins XPA, XPG, XPC, or CSB showed enhanced activation of MK2 at low doses (2.5 μM) of cisplatin compared to controls. Since XPA functions as a critical common scaffold required for both the TC-NER and GG-NER branches of the NER pathway (Fig. 1a), we examined changes in DNA damage signaling in more detail using XPA-deficient cells (Fig. 1c, XPA−). In addition, we used the same XPA-deficient cell line in which XPA function had been restored by transfection

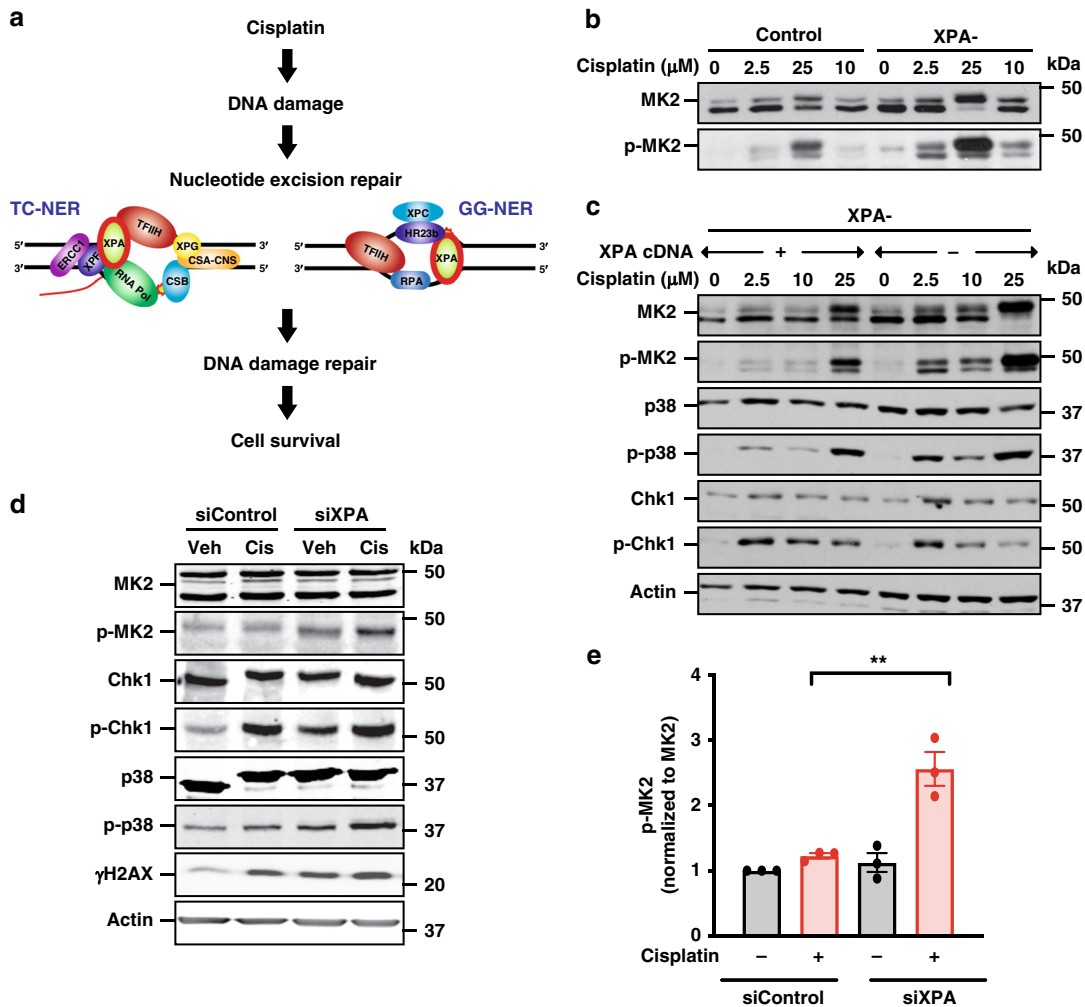

**Fig. 1 Cells defective in repair by NER hyperactivate MK2 signaling. a** Nucleotide excision repair pathways involved in cisplatin-induced DNA damage. **b** MK2 activity was assessed by Western blot analysis 24 h after vehicle or cisplatin treatment in control and XPA-deficient fibroblasts. Data are representative of two independent experiments. Note the slight activation of MK2 in the XPA deficient cells even in the absence of cisplatin treatment. **c** Western blot of MK2, Chk1, and p38 activation 24 h after cisplatin treatment in XPA-deficient fibroblasts transfected with vector control (−) or with restoration of XPA (+). **d** Western blot of MK2, Chk1, and p38 activation, and γH2AX levels, 6 h after vehicle (Veh) or cisplatin (Cis) treatment (25 μM) in XPA-proficient or deficient KP7B cells. Data representative of three independent experiments. The prominent doublet banding pattern of MK2 arises from use of alternative translation start sites in the mRNA[59]. **e** Quantification of phospho-MK2 in XPA-depleted KP7B cells. n = 3 independent experiments; **p = 0.007; two-tailed unpaired t test. Error bars represent mean ± SEM.

with the wild-type XPA gene (Fig. 1c, XPA+). Hyperactivation of both p38 (the upstream MK2 activating kinase) and MK2 observed in response to low doses of cisplatin in XPA− cells was extinguished when XPA activity was restored, indicating that increased MK2 signaling in these cells is a direct result of impaired NER activity. Interestingly, no difference in Chk1 signaling after cisplatin treatment was observed in these experiments, suggesting that these effects may be specific to MK2, rather than applicable to all checkpoint kinase signaling pathways per se (Fig. 1c).

To explore the generality of this finding, we performed similar experiments in murine K-Ras^{G12D/+}; p53^{−/−} (KP7B) lung adenocarcinoma tumor cells[31,32]. Importantly, these cells can be used in vivo with an established transplantable model of aggressive lung cancer in fully immunocompetent hosts (see below). Consistent with our XPA-MK2 findings in human fibroblasts, knockdown of XPA using siRNA in murine KP7B lung adenocarcinoma cells resulted in a similar increase in both p38 and MK2 at 6 h after treatment with cisplatin, indicating enhanced reliance on MK2 signaling in these tumor cells when

the NER pathway was impaired (Fig. 1d). Quantification of phospho-MK2 levels showed an approximately two-fold increase in the siXPA cisplatin-treated KP7B cells compared to control cisplatin-treated KP7B cells (Fig. 1e; c. f. red bars). Of note, in the lung adenocarcinoma cells in which XPA was knocked down, we observed elevated levels of γH2AX, phospho-p38, phospho-MK2, and phospho-Chk1 even in the absence of cisplatin treatment, consistent with elevated basal levels of endogenous DNA damage in tumor cells compared to non-tumorigenic fibroblasts (Fig. 1d)[33,34].

**Co-targeting NER and MK2 enhances cisplatin lethality.** The finding that NER-defective cells hyperactivate MK2 signaling, particularly in response to cisplatin-induced DNA damage suggested that these cells may have an increased reliance on MK2 to survive platinum-based chemotherapy. We therefore investigated whether co-targeting both pathways in KP7B lung adenocarcinoma cells could further enhance tumor cell death. Individual or combined knockdowns were used to determine whether loss of XPA was additive or synergistic with MK2 inhibition for tumor

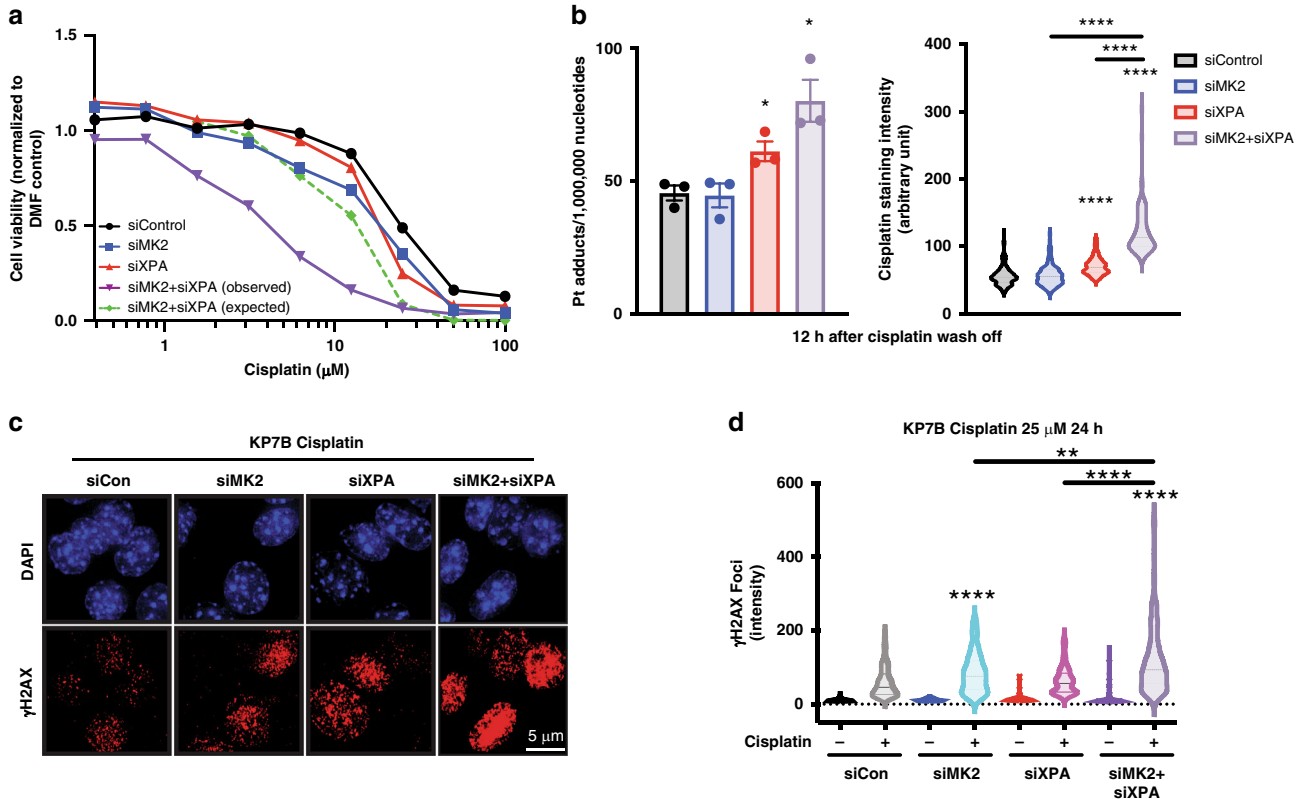

**Fig. 2 Co-targeting NER and MK2 enhances cisplatin lethality in cells. a** KP7B cells were depleted of MK2, XPA, or both using siRNA, and treated with the indicated concentrations of cisplatin. Cell viability was measured 72 h later using the CellTiter-Glo luminescence assay. The expected viability following combined MK2 and XPA knockdown was calculated assuming a Bliss independence model of additivity (see section "Methods"). **b** KP7B cells were treated with 25 μM cisplatin for 5 h. The drug-containing media was then replaced with drug-free media, and DNA repair allowed to occur for 12 h. Residual cisplatin adducts were then quantified by atomic absorption spectroscopy (left graph; siXPA + cis vs. siCon + cis, *$p = 0.0280$; siMK2 + siXPA + cis vs. siCon + cis, *$p = 0.0144$; two-tailed unpaired $t$ test. $n = 3$ experiments. Error bars represent mean ± SEM) and by immunofluorescence using an antibody against cisplatin–DNA adducts (right graph; siXPA + cis vs. siCon + cis, ****$p \leq 0.0001$; siMK2 + siXPA + cis vs. siCon + cis, ****$p \leq 0.0001$; siMK2 + siXPA + cis vs. siMK2 + cis, ****$p \leq 0.0001$; siMK2 + siXPA + cis vs. siXPA + cis, ****$p \leq 0.0001$; two-tailed unpaired $t$ test. $n = 3$ separate samples $n > 70$ cells per condition). **c** Representative immunofluorescence images of KP7B cells depleted of MK2, XPA, or both proteins treated with 25 μM cisplatin for 24 h, then fixed and stained with an antibody against γH2AX. **d** Quantification of the number of γH2AX foci in the KP7B cells treated as in Fig. 2c for each treatment in $n = 3$ separate samples. (siMK2 + cis vs siCon + cis, ****$p < 0.0001$; siMK2 + siXPA + cis vs. siCon + cis, ****$p < 0.0001$; siMK2 + siXPA + cis vs. siMK2 + cis, **$p \leq 0.002$; siMK2 + siXPA + cis vs. siXPA + cis, **$p < 0.0001$; two-tailed unpaired $t$ test. $n = 3$ separate samples and $n > 90$ cells per condition). Width in violin plot indicates frequency for each observed value from maximum to minimum, with dotted line indicating median.

cell killing by cisplatin in culture. Cells depleted of either MK2 or XPA alone showed modestly increased sensitivity to cisplatin (Fig. 2a, compare blue and red lines to black line), consistent with our prior data implicating MK2 in cell cycle arrest following DNA damage[14–17], and the ability of other DNA repair pathways such as TLS, FA, and HR to partially compensate for defective NER. Combined depletion of MK2 and XPA, however, revealed a dramatic synergistic killing of KP7B cells in response to cisplatin treatment (Fig. 2a, compare green and purple lines). Additive or greater than additive killing by cisplatin after combined MK2 and XPA knockdown, compared to individual knockdowns, was also observed in three p53 deficient NSCLC tumor cell lines (H2009, H1299, and KP7B), but not observed in a p53+ cell line (H1563) (Supplementary Fig. 2A–C). To verify that this enhanced killing was not dependent on the siRNA sequence used, a second set of siRNAs was used to co-target MK2 and XPA in KP7B with similar results (Supplementary Fig. 2D). Furthermore, to demonstrate that this effect was strictly p53-dependent, and not the result of other genetic abnormalities in the various cell lines, we next used isogenic HCT116 p53 wild type and p53 null colon cancer cells. As shown in Supplementary Fig. 2E, F, cisplatin treatment resulted in markedly reduced cell viability following combined MK2 and XPA knockdown, relative to the individual knock downs, only in the p53 null cells.

To investigate the underlying mechanism responsible for these effects, we measured platinum–DNA adducts by both atomic absorption spectroscopy on purified DNA, and by immunofluorescence of cells stained for platinum adducts. In both assays, co-depletion of MK2 and XPA greatly enhanced the number of cisplatin-DNA adducts over either knockdown alone (Fig. 2b). In addition, co-depletion of MK2 and XPA showed increased DNA damage as measured by the intensity of γH2AX foci (Fig. 2c, d), suggesting that co-targeting both MK2 and XPA in tumor cells results in a further abrogation of cisplatin–DNA adduct repair.

Together, these data suggest that MK2 and XPA have an augmented synthetic lethal relationship in p53-defective cells, and that simultaneous targeting of MK2 and XPA enhances cisplatin lethality owing to persistent DNA damage.

**siRNA–peptide nanoplex delivery to lung tumors in vivo.** Despite the therapeutic potential of MK2 inhibition to enhance the antitumor response to DNA-damaging chemotherapy, currently available small molecule MK2 inhibitors are sub-optimal and nonspecific due to the presence of a shallow ATP-binding

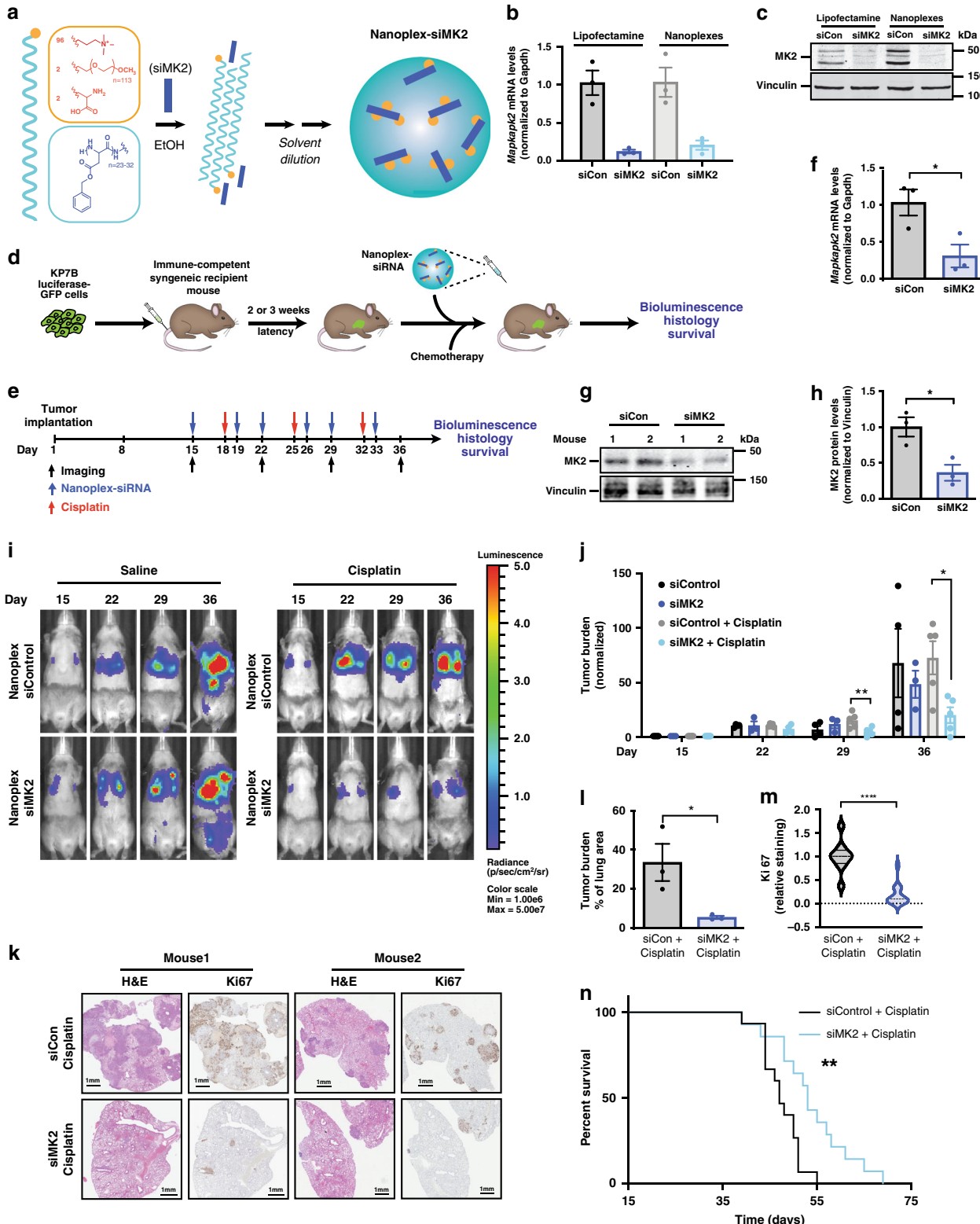

pocket in MK2. This results in cross-reactivity with MK3 and MK5, as well as several other kinases[35,36]. Furthermore, MK2 also plays an important role in the host innate immune response[37], raising the potential for systemic toxicity in response to generalized inhibition. In an effort to specifically target MK2 and co-target both MK2 and XPA within tumors, we extended a polypeptide-based nanoscale complexation approach that we have recently developed for the delivery of individual small

interfering RNAs (siRNAs) to tumors[38], in order to deliver siRNAs against MK2 (Fig. 3a) or to simultaneously co-deliver siRNAs against both MK2 and XPA (Fig. 4a). These nanoparticle delivery vehicles (RNA–peptide nanoplexes) are constructed from three unique lipid-like polypeptides containing a hydrophilic headgroup attached to a long hydrophobic alpha helical tail comprising poly(benzyl-L-aspartate) (Fig. 3a; Supplementary Fig. 3A; "Methods")[38]. Nanoplexes containing fluorescently

**Fig. 3 siRNA nanoplexes targeting MK2 improves tumor response to Pt. a** Lipid-like polypeptide nanocarriers and chemical structure of peptide-based nanoplex components. **b** Efficiency of siRNA delivery of nanoplexes in KP7B cells measured by RT-qPCR for *MK2* mRNA and **c** Western blotting for MK2 protein. Cells were harvested 72 h after lipofectamine transfection or nanoplex treatment and analyzed. Data representative of $n = 3$ experiments. **d** Schematic of nanoplex treatment in a recalcitrant syngeneic orthotopic lung adenocarcinoma mouse model. Luciferase-GFP-expressing KP7B cells form lung tumors in recipient mice, and are then treated with siRNA-loaded nanplexes 2–3 weeks later. **e** Timeline of nanoplex–siRNA and cisplatin treatment of mice with tumors. Blue arrows indicate time of nanoplex–siRNA treatment. Red arrows indicate time of cisplatin treatment. **f** Mice were sacrificed at day 36 and knockdown of MK2 mRNA measured by RT-qPCR in tumors. Data show mRNA levels as fold-change vs. nanoplex-siControl ($n = 3$ animals per group; 1 mg kg$^{-1}$ siRNA encapsulated with 200 mg kg$^{-1}$ nanoplexes; *$p = 0.0366$; two-tailed unpaired *t* test). **g** Western blot of lung tumors harvested on day 36 from 2 representative mice treated with nanoplex–siMK2 vs. nanoplexes-siCon for MK2. **h** Quantification of MK2 protein levels in tumors at day 36. Data show MK2 protein levels normalized to vinculin as fold-change vs. nanoplex–siControl. $n = 3$ animals per group, 1 mg kg$^{-1}$ nanoplex–siRNA; *$p = 0.0215$; two-tailed unpaired *t* test. **i** Representative bioluminescence images before and after nanoplex–siRNA and saline or cisplatin treatment. **j** Quantification of lung bioluminescence following first, second, and third treatments after tumor induction, shown as fold-change compared to pre-treatment (day 15) (nanoplex–siCon $n = 4$ animals, nanoplex–siMK2 $n = 3$ animals, nanoplex–siCon + cisplatin $n = 5$ animals, nanoplex–siMK2 + cisplatin $n = 5$ animals; *$p \leq 0.0143$; two-tailed unpaired *t* test, post third treatment). **k** Representative H&E and Ki-67 staining of lungs at the end of three rounds of treatment. **l** Quantification of tumor burden as a percentage of lung area. $n = 3$ animals per condition; *$p = 0.0422$; two-tailed unpaired *t* test. **m** Quantification of Ki-67 as a percentage of positive cells. $n = 3$ animals per condition; ****$p \leq 0.0001$; two-tailed unpaired *t* test. Data shown as violin plots as in Fig. 2d. **n** Kaplan–Meier survival analysis of tumor-bearing mice treated with nanoplex–siMK2 or nanoplex–siCon and cisplatin. $n = 3$ animals per condition; *$p = 0.0035$, calculated using the log-rank test. Error bars in panels **b**, **f**, **h**, **j**, and **l** represent mean ± SEM.

tagged siRNAs form punctate foci in cells at early time points, consistent with endosomal/lysosomal cellular uptake (Supplementary Fig. 3B). These steps are followed by gradual endosomal escape and release of siRNA to the cytoplasm[38]. When the nanoplexes were used to deliver siRNA to cells in vitro, single and combined target knockdown efficiencies at least as effective as those obtained using conventional cationic lipids were observed (Fig. 3b, c), with little indiscriminant toxicity at varying siRNA and nanoplex concentrations and ratios (Supplementary Fig. 3C). Nanoplexes efficiently delivered siRNA against MK2 and XPA and knocked down both genes at the single cell level, as shown by immunofluorescence staining in KP7B cells treated with nanoplex–siMK2, nanoplex–siXPA, and nanoplex–siMK2/siXPA (Supplementary Fig. 3D–F). We found using immunofluorescence that >95% of the KP7B cells treated with nanoplex–siMK2/siXPA show dual MK2 and XPA knockdown (Supplementary Fig. 3F). Interestingly, using an ex vivo flow cytometry based assay of tumor cells cocultured with splenocytes, we observed that the siRNA nanoplexes were preferentially taken up by the tumor cells, compared to macrophages or T-cells (Supplementary Fig. 3G)[39], suggesting that this nanoparticle approach might limit the effect of MK2 inhibition in immune cells.

To determine whether the nanoplexes could be used to deliver single or dual-targeting siRNAs to lung adenocarcinomas in vivo, we took advantage of an aggressive transplantable model where tumors are generated in immunocompetent, syngeneic recipient animals by tail vein injection of *K-Ras*$^{G12D/+}$; *p53*$^{-/-}$ (KP7B) tumor cells[31,32], followed by subsequent tumor seeding in the lung (Fig. 3d). We choose to use this model for our experiments because the transplanted cells give rise to tumors in mice with fully functional immune systems, at the correct anatomical location, and result in murine tumors that are pathologically and molecularly similar to the tumors from which they were derived[31], as well as their human tumor counterparts[40,41].

As a control for nanoparticle delivery of siRNAs to lung adenocarcinomas in vivo, nanoplexes containing a fluorescently tagged control non-targeting siRNA were administered intraperitoneally to mice with luciferase/GFP-expressing lung tumors (Fig. 3d; Supplementary Fig. 4A). The siRNA nanoplexes localized to the vicinity of the tumors and delivered ~5% of the initial siRNA injected dose as determined by co-registry of lung tumor bioluminescence with thoracic tomograms of fluorescently labeled siRNA (Supplementary Fig. 4B, C). By comparison, small molecule inhibitors typically deliver <1% of the initial drug dose

to the tumor[42,43]. Removal of nanoparticles from the body typically involves a combination of hepatic and renal clearance[44]. We observed no hepatic or renal dysfunction as determined by serum biochemical markers 24 h after treatment (Supplementary Fig. 4D–I). To ensure that the nanoplexes that we engineered are not immunogenic in this fully immunocompetent mouse model, serum cytokine profiles were measured 24 h after nanoplex/siRNA delivery. There were minimal changes in the levels of serum cytokines in nanoplex-treated animals compared to controls (Supplementary Fig. 4J). In particular, the levels of IL-6 and TNFα, classic indicators of an innate immunogenic response, were unchanged. These data suggest that our nanoplex carriers can efficiently deliver siRNAs to lung adenocarcinoma tumors in vivo.

**siRNA nanoplex targeting of MK2 improves Pt response in NSCLC.** We next used the nanoplexes to specifically target MK2 alone in vivo in pre-existing lung tumors using the aforementioned immunocompetent transplant model. As shown in Fig. 3e, tumors were allowed to form for 14 days prior to treatment in order to assess therapeutic response in established tumors. Mice were then treated with non-targeting (control) or MK2-targeting siRNA-loaded nanoplexes (1 mg kg$^{-1}$) twice weekly on days 1 and 4, followed by a single dose of cisplatin (7 mg kg$^{-1}$) on day 3 of each week for a total of 3 weekly cycles (Fig. 3e). The administration of systemic cisplatin was chosen to correspond with the time when maximal mRNA and protein inhibition was observed in in vitro pilot studies (Supplementary Fig. 5). A second nanoplex–siRNA dose was given at day 4 to ensure sustained knockdown. Analysis of MK2 mRNA and protein levels from tumors excised at day 36, performed at the conclusion of the experiment, validated that the MK2 siRNA nanoplexes had significantly reduced the levels of MK2 mRNA and protein in the tumors by 70% (Fig. 3f) and 65% (Fig. 3g, h), respectively, in vivo.

Both control and MK2 siRNA nanoplex-treated animals showed a similar level of tumor bioluminescence in the absence of DNA damaging chemotherapy (Fig. 3i, j, compare black and dark blue bars), indicating that depletion of MK2 alone had no significant effect on tumor growth in this model. Following systemic administration (intraperitoneal injection, IP) of cisplatin, control tumors continued to grow rapidly despite platinum treatment, (Fig. 3i, j, compare black and gray bars). In contrast, cisplatin treatment markedly inhibited the growth of MK2-depleted tumors (Fig. 3i, j, compare gray and cyan bars). By the conclusion of the experiment at day 36, the tumor burden in mice

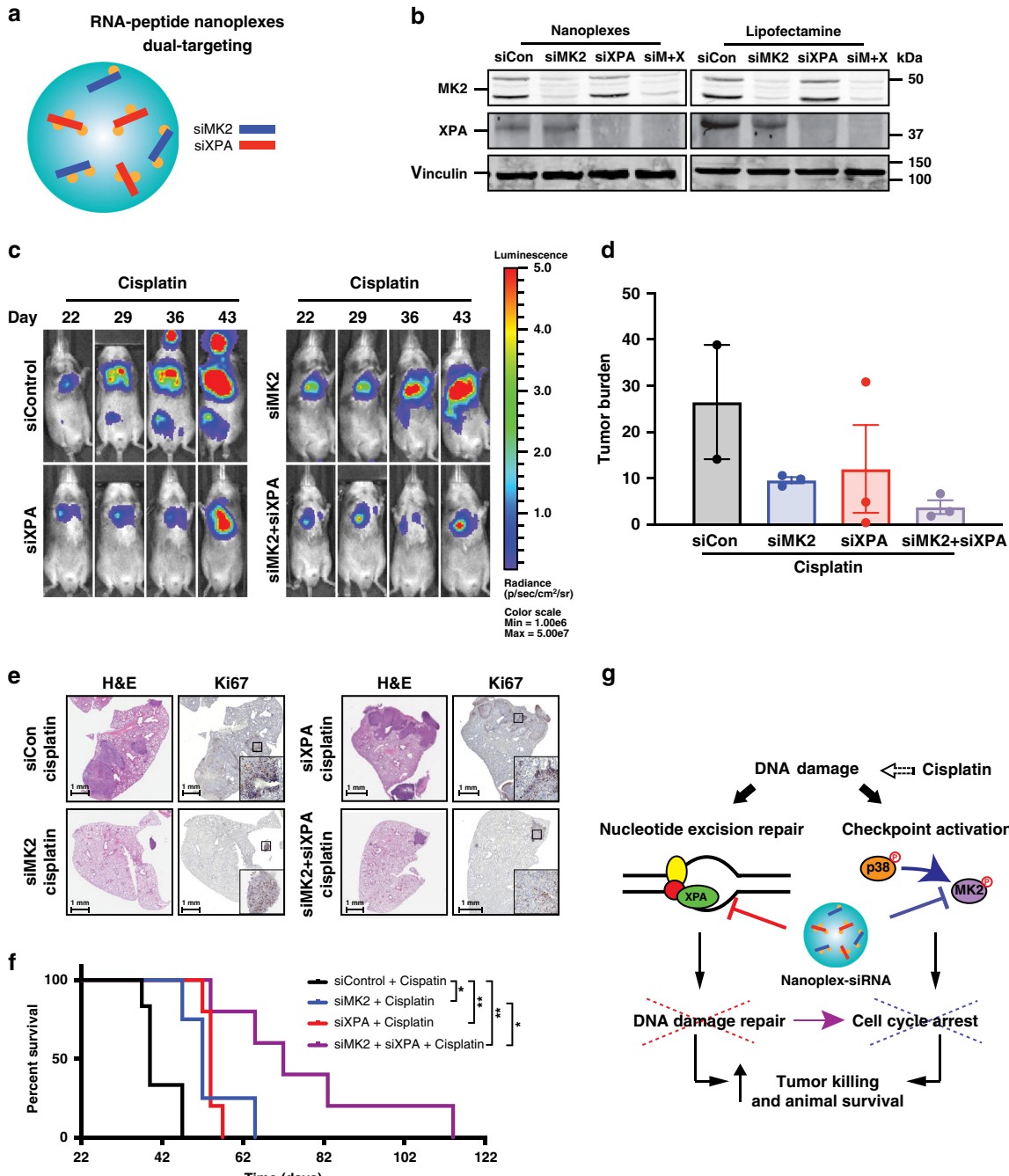

**Fig. 4 Augmented synthetic lethality for Pt by co-targeting XPA and MK2 in vivo. a** Schematic of dual-targeting peptide-based nanoplexes. **b** MK2 and XPA knockdown efficiency of nanoplex–siMK2/siXPA compared to lipofectamine RNAiMax–siMK2/siXPA measured by Western blotting for MK2 and XPA. Data representative of two independent experiments. **c** Representative bioluminescence images before and after indicated siRNA and cisplatin treatment on days 22, 29, 36, and 43. **d** Quantification of lung bioluminescence at 43 days after tumor implantation ($n = 3$ animals were used for each condition; only two mice remained alive in the nanoplex–siControl + cisplatin at day 43, and they died by day 50). Error bars represent mean ± SEM. **e** Representative H&E and Ki67 lung staining at the end of three rounds of the indicated treatments. Three animals were used for each condition. **f** Kaplan–Meier survival analysis of tumor-bearing mice treated with the indicated nanoplex–siRNA in combination with cisplatin treatment (nanoplex–siControl + cisplatin $n = 6$ animals, nanoplex–siMK2 + cisplatin $n = 3$ animals, nanoplex–siXPA + cisplatin $n = 5$ animals, and nanoplex–siMK2/siXPA + cisplatin $n = 5$ animals. *$p \leq 0.05$ and **$p \leq 0.01$ calculated using the log-rank test). **g** Model illustrating crosstalk between the MK2 signaling pathway and nucleotide excision repair in p53-defective cells. Co-targeting these pathways in established tumors prolongs spontaneous survival and potently enhances the antitumor response to cisplatin treatment.

with established tumors treated in vivo with siMK2–nanoplexes in combination with cisplatin was 3.5-fold lower than that of the controls (Fig. 3j). These data demonstrate, for the first time, superior responses to cisplatin-based chemotherapy in established MK2-proficient tumors subsequently depleted of MK2 by a drug-like modality. The increased efficacy of cisplatin in lung tumors depleted of MK2 was further validated by histology and immunohistochemistry (Fig. 3k). Quantification of H&E staining of lung sections from cisplatin-treated animals who received nanoplex-siMK2 demonstrated a six-fold reduction in tumor

burden and Ki67 staining compared to control siRNA nanoplex-treated animals (Fig. 3l, m), indicating reduced proliferation in tumors treated with the combination of nanoplex–siMK2 and cisplatin. The observed increase in platinum response seen in established lung tumors upon MK2 knockdown with nanoplex-targeted siRNA in vivo further resulted in a modest but significant extension of lifespan following platinum therapy (Fig. 3n). We confirmed that our nanoplex–siMK2 results were due to inhibition of tumor cell MK2, and not due to off-target effects, by examining the response of an independent set of mice. These animals bore lung tumors derived from KP7B cells expressing a MK2-targetting shRNA that targets an alternative sequence within the MK2 mRNA with remarkably similar results (Supplementary Fig. 6A–F). Together, these data provide compelling evidence that inhibition of MK2 by therapeutic RNAi nanoparticles can markedly augment the response of established, aggressive tumors to frontline platinum-based chemotherapy and improve survival.

**Loss of XPA synergizes with MK2 inhibition in NSCLC tumors.** Although MK2 depletion clearly enhanced tumor cell responses to cisplatin (Fig. 3i–m; Supplementary Fig. 6D, E), the overall increase in survival was modest (Fig. 3n; Supplementary Fig. 6F). We next investigated whether we could exploit the ASL relationship between XPA and MK2 (Fig. 2) for treatment of murine lung adenocarcinomas using our nanoplexes to co-deliver siRNAs against both MK2 and XPA in vivo. Dual targeting nanoplex–siRNA delivery vehicles against MK2 and XPA were synthesized and validated for their ability to simultaneously knockdown both MK2 and XPA (Fig. 4a, b). Using the immunocompetent murine lung adenocarcinoma transplant model described in Fig. 3d, tumors were allowed to form, but treatment was withheld until 21 days post transplantation (Supplementary Fig. 7A). This period is 1-week longer than that shown in Fig. 3, in order to allow the tumors to grow even larger, thus raising the bar for therapeutic intervention. Mice were then treated with non-targeting nanoplex–siRNA (control), nanoplex–siRNA against MK2 alone or XPA alone, or with dual-targeted nanoplex–siRNA directed against both MK2 and XPA, on a twice weekly schedule, along with administration of a single dose of cisplatin each week for 3 weeks. Of note, both tumor-bearing and nontumor-bearing mice treated with nanoplex-siRNA against MK2 and XPA in combination with cisplatin did not have excessive weight loss compared to control mice treated with cisplatin alone, consistent with minimal direct toxicity of the nanoparticle formulation even when administered in combination with a platinum agent (Supplementary Fig. 7B, C). Cisplatin treatment markedly suppressed the growth of MK2, XPA, or MK2/XPA-depleted tumors compared to controls (Fig. 4c, d). At day 36, the tumor burden of mice treated with nanoplex-siXPA was 2.8-fold less (Supplementary Fig. 8A, red vs. black post-treatment bars), and nanoplex–siMK2/siXPA was 20-fold less, than those of the controls treated with cisplatin alone (Supplementary Fig. 8A, purple vs. black bars). Furthermore, the tumor burden of mice treated with dual-targeted nanoplex–siMK2/siXPA was threefold less than that of the animals treated with nanoplex–siMK2 alone (Supplementary Fig. 8A, purple vs. blue post treatment bars) at this same time point. By day 43 (22 days post initial treatment), we noted that mice receiving nanoplex-siMK2/siXPA targeted combination continued to sustain lower tumor burden even after treatment cessation on day 33 (Fig. 4d). Both H&E and Ki67 staining performed on lung sections from mice that either received nanoplex–siMK2, nanoplex–siXPA, or the nanoplex–siMK2/siXPA combination with cisplatin further confirmed reductions in tumor burden and proliferation (Fig. 4e),

consistent with the results of whole animal bioluminescence imaging (Fig. 4c, d). Together, these data indicate that, whereas MK2 and XPA inhibition alone sensitizes established refractory lung adenocarcinoma tumors to cisplatin, (with MK2 inhibition being superior to that of XPA inhibition alone, based on residual tumor burden), co-targeting of XPA and MK2 within the same tumor further reduces tumor burden with a notably increased efficiency (Fig. 4c–e; Supplementary Fig. 8A, B).

To determine whether the observed response to cisplatin in tumors co-depleted of MK2/XPA translated into a significant overall therapeutic benefit to the animals compared to MK2 depletion alone, we analyzed long-term animal survival. Mice that received either single nanoplex–siXPA or nanoplex–siMK2 treatment in combination with cisplatin displayed a significant but modest survival benefit (Fig. 4f; median survival 32 and 30 days, respectively, compared to 17 days for nanoplex–siControl). However, co-depletion of both XPA and MK2 using dual targeted XPA/MK2 siRNA nanoplexes in combination with cisplatin had a strikingly profound effect, increasing the median survival to 50 days, and more than doubling the survival of longest-lived animals in any of the other cohorts (Fig. 4f). These data indicate establishment of ASL in vivo by combined MK2/XPA depletion significantly increases the overall therapeutic benefit of cisplatin and is superior compared to depletion of MK2 or XPA alone.

Remarkably, we also observed an unexpected significant survival benefit from long-term simultaneous co-targeting of XPA and MK2 within tumors even in the absence of cisplatin treatment (Supplementary Fig. 8C), consistent with the previously observed increase in spontaneous DNA damage signaling observed in XPA depleted cells in culture (Fig. 1b–d). The addition of cisplatin, however, further extended the median survival of animals treated with combination MK2/XPA siRNA nanoplexes by an additional 15 days (Supplementary Fig. 8C). Taken together, these data indicate that the combined targeting of DNA damage-induced cell cycle checkpoint pathways and DNA repair via co-inhibition of MK2 and XPA, respectively using polypeptide-based nanocarriers results in spontaneous DNA damage and antitumor responses, that can be further enhanced by the addition of cisplatin to improve therapeutic outcomes in established cisplatin-resistant lung tumors in vivo (Fig. 4g).

## Discussion

Manipulation of the DNA-damage response, either by disrupting cell cycle checkpoints, or by interfering with DNA repair, is emerging as a promising approach to promote tumor cell killing and enhance the response to DNA-damaging chemotherapeutic drugs[4]. Three checkpoint effector kinases, Chk1, Chk2, and MK2, play critical roles in initiating or maintaining cell cycle arrest after genotoxic damage[45]. Chk1 activity in the nucleus is required for the initiation of cell cycle arrest after DNA damage[16], and small molecule inhibitors for systemic administration are currently under investigation in Phase 1 and 2 clinical trials[4]. Although promising, their use has been complicated by significant dose-limiting toxicity[4,46]. Since Chk1 function is critical during normal DNA replication, homozygous genetic deletion of *Chk1* results in embryonic lethality, while heterozygous *Chk1*+/− animals have a significantly increased propensity to develop cancer[47]. Prolonged inhibition of Chk1 function in non-tumorigenic tissues during cancer therapy therefore also inadvertently increase the incidence of secondary malignancies.

In contrast, mice with homozygous and heterozygous loss of *MK2* are fully viable[37], and the essential function of MK2 as a checkpoint kinase become prominently unmasked, in a synthetic lethal manner, when p53 function is lost, making it an ideal target

for certain anticancer therapies[14]. MK2, when activated downstream of ATM and ATR, acts within the cytoplasm, independently of CHK1, to maintain G1/S, and intra-S cell cycle checkpoints through p27, and the G2/M checkpoint through Gadd45[14,16,18,48]. To date, however, there are no MK2 inhibitors approved for clinical use[4], and the small molecule inhibitors that are currently available for laboratory and animal use are both sub-optimal and non-specific, owing to the shallow MK2 ATP-binding pocket[35,36]. In addition, MK2 has a well-established role in innate immunity and inflammatory signaling[37], similar to DNA-PK, raising the possibility of off-target side effects from systemic inhibition. To overcome these limitations, we investigated a tumor-targeting nanoparticle that delivers siRNAs to the site of pre-existing MK2-containing tumors and efficiently depletes both its target RNA and protein.

In addition to targeting cell cycle checkpoint pathways, the specific targeting of DNA repair pathways to improve the efficacy of genotoxic chemotherapy may be particularly important in NSCLC, since evidence suggests that these tumors have a pre-existing increase in intrinsic DNA repair activity, possibly as a consequence of long-time adaptation to DNA lesions induced by cigarette smoking[49,50]. We focused on the NER pathway, which is required for the repair of cisplatin adducts. XPA depletion alone resulted in elevated levels of MK2 signaling which could be further enhanced by exogenous genotoxic injury with cisplatin. Furthermore, this combined MK2 and XPA loss resulted in increased platinum-induced DNA lesions. These results indicate that, in the absence of a functional NER pathway, MK2 appears to be hyperactivated to assist in cell cycle arrest and damage repair. Loss of XPA synergistically augmented the pre-existing dependency of p53-defective lung adenocarcinoma cells on MK2 to survive cisplatin treatment in vitro, revealing an augmented synthetic lethal relationship between p53, MK2, and XPA. This mammalian cell trigenic interaction which we unmasked upon cisplatin treatment, bears some conceptual similarity to a trigenic mutant screening and colony size scoring approach performed in budding yeast by Kuzmin et al.[51], though in that work extrinsic perturbations such as genotoxic stress, were not used.

To implement this ASL relationship between p53, MK2, and XPA in the setting of platinum-induced DNA damage in vivo, we co-encapsulated siRNAs against MK2 and XPA into the same nanoplex particles. These were then administered alone or in conjunction with systemic cisplatin treatment, resulting in a markedly improved anti-tumor response of established lung tumors in vivo upon platinum treatment that was associated with a dramatically prolonged survival. In vivo, tumor cell-enriched nanoparticle uptake is likely to be further exploited by the preferential targeting of the nanoparticles to tumors as a consequence of so-called "passive" targeting[52]. Nanoparticles are able to exploit the cancer's distinct vascular and lymphatic pathology (i.e., leaky vasculature and defective lymphatic drainage) to accumulate in the tumor. Here we utilize the enhanced tumor cell uptake and passive tumor targeting properties to formulate nanoparticles that are able to pass through the leaky vascular junctions in tumors resulting in preferential accumulation at the tumor site over time, as shown in Supplementary Fig. 4. The fact that not all of the tumor cells were killed in vivo suggests that not all tumor cells may have internalized the nanoparticles, consistent with the fractional uptake that we observed in vitro. Design strategies to improve uptake of the nanoplexes within tumors might result in even better tumor cell killing.

To our knowledge, this study is the first to simultaneously target both a DNA repair pathway and a cell cycle checkpoint pathway with a drug-like modality in vivo, both alone and in the context of cytotoxic chemotherapy treatment. Taken together, our data indicate that: (1) cross-talk exists between the NER and p38/MK2 pathways, which coordinates DNA repair and cell fate after DNA damage, and that (2) this synergistic interplay between two key biological pathways for DNA damage response can result in highly effective targeted combination therapies for cancer treatment. Overall these data establish the paradigm that identification and therapeutic targeting of augmented synthetic lethal relationships can produce a safe and highly effective therapy by 're-wiring' multiple DNA damage response pathways, the systemic inhibition might otherwise be toxic. Our nanoplexes are highly modular, and consequently they facilitate the encapsulation of any combination of siRNAs, leading to rapid translation of defined SL or ASL interactions from discovery to therapeutic targeting in vivo.

## Methods

**Cell culture**. All human cell lines were purchased from ATCC (American Type Culture Collection) or Coriell Institute. HCT116 p53 Null cells were a gift from the Vogelstein laboratory. H1299 and H1563 cells were grown in RPMI supplemented with 10% fetal bovine serum (FBS) and 2 mM L-Glutamine. 293T, H2009 and KP7B cells were grown in DMEM supplemented with 10% FBS and 2 mM L-Glutamine. KP7B cells were described previously by Doles et al.[31], and were a gift from Tyler Jacks' laboratory. HCT116 cells were grown in McCoy's 5a supplemented with 10% FBS and 2 mM L-Glutamine. Human fibroblast cell lines GM15876A and GM04312 were grown in DMEM supplemented with 10% FBS and 2 mM L-Glutamine. Human non-transformed fibroblast cell lines GM16684, GM00739, and GM16181 were grown in DMEM supplemented with 15% FBS and 2 mM L-Glutamine. All cell lines were cultured in a 37 °C humidified incubator with 5% $CO_2$, maintained subconfluently and used for no more than 20 passages.

**Antibodies and chemicals**. Antibodies against MK2/MAPKAPK2 (#3042, 1:1000), Phospho-Thr-334 MAPKAPK-2 (#3041), Phospho-Ser345 Chk1 (#2348, 1:1000), Phospho-Thr-180/Tyr-182 p38 (#9211, 1:1000), Phospho-Ser-139 Histone H2AX (#9718, 1:1000), and Histone H2AX (D17A3, #7631, 1:1000), were purchased from Cell Signaling Technologies. Antibodies for Chk1 (#8408, 1:1000), p38 (A-12, sc-7972, 1:1000) and XPA (B-1 sc-28353, 1:500) were purchased from Santa Cruz biotechnology. Ki-67 (ab16667, 1:1000), and anti-cisplatin modified DNA (CP9/19; Abcam ab103261, 1:500) antibodies were purchased from Abcam. Antibodies against β-Actin (A5441, 1:2000), GAPDH (G8795, 1:5000) and vinculin (V4505, 1:5000), as well as doxorubicin and cisplatin were purchased from Sigma Aldrich. All chemicals were used at the indicated doses. Antibodies for flow cytometry, CD45-PECy7 (Clone: 30-F11, #25-0451-82, 1:100), CD11b-FITC (Clone: M1/70, #11-0112082, 1:100), and F480-PE (Clone: BM8, #12-4801-82, 1:100) were from Thermo (eBioscience) while CD3e-APC-Cy7 (Clone: 145-2C11, #100330, 1:100) was from BioLegend.

**Retro-virus production**. For VSVG-pseudotyped virus production, 293T cells were transfected using the calcium phosphate method (Clontech) using either pMLS-tomato (for shRNAs in KP7B cells) along with packaging and structural vectors VSVG and GAG/POL. Supernatants containing virus were then used to transduce target cells in the presence of 8 μg/mL polybrene for three rounds of infection. Successfully transduced cells were sorted for tdTomato expression by flow cytometry for pMLS-tomato infected cells. GFP-Luciferase was used to infect KP7B cells for bioluminescent imaging and was a kind gift from Bonnie Huang (MIT).

**shRNA sequences**. All shRNAs were designed using the Cold Spring Harbor web portal (http://katahdin.cshl.org/siRNA/RNAi.cgi?type=shRNA) and 97mer oligo-nucleotides (see below for sequence, underlined sequences are gene-specific) were used as templates for PCR using miR-30 shRNA amplification primers (see below). PCR products were digested with XhoI and EcoRI and ligated into pMLS-Tomato.
shMK2 mmu
TGCTGTTGACAGTGAGCG<u>ATCCTTGGGTGTCATCATGTATT</u>AGTGAA GCCACAGATGTA<u>ATACATGATGACACCCAAGGACT</u>GCCTACTGCCTCG GA
miR-30 cloning XhoI Fwd primer
CAGAAGGCTCGAGAAGGTATATTGCTGTTGACAGTGAGCG
miR-30 cloning EcoRI Rev primer
CTAAAGTAGCCCCTTGAATTCCGAGGCAGTAGGCA

**siRNA oligonucleotides and siRNA transfection**. MK2, XPA siRNA, and non-targeting control siRNA (Silencer negative control No. 1 siRNA) were purchased from Ambion. siRNA transfection was performed using Lipofectamine RNAiMAX as per manufacturer's instruction (Invitrogen) using a final concentration of 5 nM siRNA in H1299 and H2009 and 50 nM in KP7B cells unless otherwise stated.
Sequences of Silencer select siRNA are as follows: Murine MAPKAPK2 (s201671) sense: ACAGAAUUCAUGAACCACCTT, antisense: GGUGGUUCA UGAAUUCUGUGA; murine MAPKAPK2-2 (s201670) sense: GAACGAUGG

GAGGAUGUCATT, antisense: UGACAUCCUCCCAUCGUUCCT; murine XPA (s76138) sense: GCUUAUAACCAAGACAGAATT, antisense: UUCUGUCCUU GGUUAUAAGCTT; murine XPA-2 (s76140) sense: CCAAAAUGAUUGACACCAATT, antisense: UUGGUGUCAAUCAUUUUGG GA; human MK2 (s569) sense: GGAUCAUGCAAUCAACAAATT, antisense: UU UGUUGAUUGCAUGAUCCAA; and human XPA (s14925) sense: GAAGAUGA CAUGUACCGUATT, antisense: UACGGUACAUGUCAUCUUCTA. Fluorescently labeled siRNA was purchased from Qiagen (AllStars Negative Control siRNA, Alexa Fluor 647; Qiagen).

**Electron microscopy.** TEM was performed using a JEOL 2100 FEG instrument equipped with a Gatan 626 Single Tilt Liquid Nitrogen Cryo Transfer Holder. Samples were prepared on QUANTIFOIL Holey Carbon Films (Electron Microscopy Sciences) using a Gatan Cryo-Plunge3 system.

**Survival assays.** Survival assays were performed on either 96-well (2500 cells) or 384-well (500 cells) plates. Cells were transfected with siRNA as described above and treated with various doses of cisplatin 48 h after transfection. Cell viability was measured 72 h after cisplatin treatment using the CellTiter-Glo luminescent cell viability assay as per manufacturer's instructions (Promega) using the Tecan Infinite 200 PRO plate reader. Expected values were calculated based on the Bliss independence model of additivity following methods outlined in Foucquier and Guedj[53], (siRNA1 viability × siRNA2 viability × cisplatin viability).

**In vivo biodistribution.** Tumor accumulation was approximated using a Xenogen IVIS Imaging System (Caliper). Tumor-bearing mice ($n = 3$) were intraperitoneally injected with nanoparticles (1 mg kg$^{-1}$, AllStars Negative Control siRNA, Alexa Fluor 647; Qiagen) dispersed in phosphate-buffered saline (PBS). After 24 h, mice were anesthetized with isoflurane and imaged (640/700 nm ex/em) using aperture-limited transillumination fluorescence tomography. The signal in tomographic images was spatially limited to that from the upper abdomen. Biodistribution measurements were taken from whole animal epifluorescence images. Recovered fluorescence from the lungs and whole animal were quantified using the region-of-interest analysis package in Living Image (Perkin Elmer).

**Analysis of differential siRNA uptake by tumor and immune cells using splenocyte–tumor cell coculture.** Spleens were isolated from C57BL6/Jx129-JAE mice and mashed through a 40 μm filter. Red blood cells were lysed after incubation with ACK lysis buffer for 5 min and splenocytes were washed with complete growth media (RPMI, 10% FBS, 20 mM HEPES, 1 mM sodium pyruvate, 0.055 mM 2-mercaptoethanol, 2 mM L-glutamine, 1× nonessential amino acids and antibiotics). 500,000 splenocytes were co-cultured with 500,000 KP7B tumor cells per replicate per condition and treated for 24 h with AF647-siRNA-nanoparticles as indicated. Viable cells were assessed by flow cytometry for uptake of fluorescent siRNA by specific immune populations by co-staining with fluorophore conjugated antibodies for CD45, CD11b, F480, and CD3. CD45+ CD11b+ AF647+ cells were scored as siRNA+ macrophages, CD45+ CD3+ AF647+ cells were scored as siRNA+ T-cells and CD45-AF647+ cells were scored as siRNA+ tumor cells. The gating strategy is shown in Supplementary Fig. 9.

**Fluorescence microscopy.** Live-cell fluorescence imaging was performed using a Nikon 1AR Ultra-Fast Spectral Scanning Confocal Microscope equipped with an environmental chamber providing temperature control. KP7B cells were passaged onto 35 mm glass-bottom culture dishes (MatTek). Adherent cells were washed with DPBS and concurrently incubated with nanoparticles (50 nM, BLOCK-iT Red control siRNA, Thermo Fisher), LysoTracker Deep Red (50 nM, Thermo Fisher), and Hoechst 34580 (10 ug/mL, Thermo Fisher) in Opti-MEM media at 37 °C for 1 h. Cell monolayers were then washed in DPBS and imaged in HEPES buffer (10 mM, pH 7.4). Immunofluorescence was performed by seeding cells onto poly-L-Lysine coated glass coverslips in 12-well plates and treated with cisplatin. Cells were washed in PBS, and then fixed with a 4% paraformaldehyde solution for 20 min at room temperature. Cells were blocked in 16.6% goat serum, 0.3% Triton X-100, 20 mM sodium phosphate, and 0.45 M sodium chloride prior to incubation in primary antibody overnight followed by secondary antibodies conjugated with Alexa Fluor (Invitrogen). Coverslips were then mounted using ProLong Gold Antifade Mountant (Invitrogen P36934) and imaged on Nikon Eclipse 800i fluorescence microscope and Applied Precision DeltaVision Ultimate Focus Micropscope with TIRF Module. CellProfiler 3.0.0 (developed by the Broad Institute of MIT and Harvard's Imaging Platform and available at http://www.cellprofiler.org) was used to quantify the integrated intensity of cells stained with γH2AX and cisplatin-DNA adduct antibodies. Each image was broken down into its component grayscale images for analysis using a CellProfiler pipeline. An illumination function module was utilized to uniformly reduce and smoothen background staining on the entire image set. The cisplatin–DNA adduct antibody images were further processed with a module to reduce background signal outside the nucleus. Cell nuclei were defined as primary objects using the DAPI grayscale images. The integrated intensity within the primary objects was then quantified for cisplatin adducts and γH2AX foci. Quantification of cells staining positively for

MK2 and XPA, using a DAPI overlay, was performed using ImageJ Fuji 1.0. The same thresholding values were applied to all images.

**Cell toxicity studies.** Cell toxicity studies were performed at various N:P ratios (the ratio of positively charged polymer amine to negatively charged nucleic acid (siRNA) and assayed using the CellTiter-Glo luminescent cell viability assay (Promega).

**In vivo cytokine and toxicity studies.** Serum and plasma from mice treated with nanoplex-siRNA (200 mg kg$^{-1}$ nanoplexes and 1 mg kg$^{-1}$ siRNA) obtained at 24 h following treatment. Albumin, blood urea nitrogen (BUN), and creatinine (Cr) levels were measured by Charles River Laboratories from serum samples obtained via cardiac puncture 24 h following nanoplex–siRNA administration in 8- to 10-week-old male C57BL6/Jx129-JAE mice. Cytokine levels in plasma were measured by Eve Technologies and visualized using TIGR MeV version 4.9.

**Histology.** Tumor samples were obtained at 72 h following final nanoplex-siRNA administration, then formalin-fixed, paraffin-embedded, stained (H&E and Ki67), and scanned (Aperio slide scanner, Leica Biosystems).

**qRT-PCR.** For qRT-PCR analysis RNA was extracted from mouse tumor or cells using TRIzol reagent (Ambion) according to the manufacturer's instructions and 1 μg of total RNA was used for reverse transcription using the superscript III first-strand synthesis kit (Invitrogen) as per the manufacturer's instructions. For qPCR cDNA was amplified using SYBR green PCR mastermix (Applied Biosystems) according to the manufacturer's cycling conditions for 40 cycles on a Bio-Rad C1000 Thermal Cycler. Data were analyzed using the delta-delta Ct method and plotted as fold change vs. control[15]. Primers were ordered from Invitrogen and/or IDT: Actin *mmu* Fwd TGTTACCAACTGGGACGACA, Actin *mmu* Rev GGGG TGTTGAAGGTCTCAAA, Gapdh *mmu* Fwd GGGAAATTCAACGGCACAGT, Gapdh *mmu* Rev AGATGGTGATGGGCTTCCC, MK2 *mmu* Fwd CTTCCAAA AGGCCCAATGCC, MK2 *mmu* Rev GGACTTCCGGAGCCACATAG, XPA *mmu* Fwd ACTGCTTCTTATTGCTCGCC, and XPA *mmu* Rev AGCTCTGGAAGATGCAAAGG.

**Immunoblot analysis.** Cells were lysed in RIPA lysis buffer containing protease and phosphatase inhibitor (Roche). Protein concentration was measured using BCA (Pierce). Cell extracts containing the same amount of protein in every condition were mixed with 6× reducing sample buffer and boiled at 95 °C for 5 min, and subjected to electrophoresis using the standard sodium dodecyl sulfate polyacrylamide gel electrophoresis method. For LICOR-based blotting, proteins were transferred to nitrocellulose membranes (Biorad; Catalog# 162-0115) and blocked with Odyssey blocking buffer for 1 h. Primary antibodies were then incubated overnight at 4 °C followed by secondary antibodies conjugated with LICOR fluorophores. Samples were scanned with a LICOR/Odyssey infrared imaging system (LICOR Biosciences) and band densitometries were quantified using ImageStudio. For enhanced chemiluminescence (ECL)-based blotting, proteins were transferred to methanol-activated PVDF membranes (Biorad; Catalog# 162-0177) and blocked with 5% nonfat dried milk for 1 h. Primary antibodies were then incubated overnight at 4 °C followed by secondary antibodies conjugated with HRP for developing with ECL (Perkin Elmer). Uncropped versions of blots are included in Supplementary Figs. 10 and 11, and also provided as Source Data File.

**Murine lung adenocarcinoma transplant model.** KP7B cells ($5 \times 10^4$), labeled with GFP-Luciferase, were transplanted into 10- to 12-week-old syngeneic C57BL6/Jx129-JAE male recipient mice 6 h after 5 Gy whole body irradiation[18,31]. Tumors were allowed to form for 2-3 weeks and tumor growth was measured by bioluminescent imaging. Mice were injected IP with 165 mg kg$^{-1}$ luciferin 10 min prior to bioluminescent imaging on an IVIS Spectrum-bioluminescence and fluorescent imaging system (Xenogen Corporation). For all imaging procedures animals were pre-anesthetized with isoflourane. For siRNA treatment, nanoplex-siRNA (200 mg kg$^{-1}$ nanoplexes, 1 mg kg$^{-1}$ siRNA) was injected twice weekly for 3 weeks. For drug treatments, cisplatin was dissolved in saline and injected IP at 15 mg kg$^{-1}$ for single high dose treatment, or with 7 mg kg$^{-1}$ once weekly for a total of three weeks for low dose treatment. Mice were sacrificed when moribund or when they had lost 20% of their initial body weight, whichever occurred sooner, according to MIT Committee on Animal Care guidelines. All mouse studies were approved by the MIT Institutional Committee for Animal Care (CAC), and conducted in compliance with the Animal Welfare Act Regulations and other federal statutes relating to animals and experiments involving animals and adheres to the principles set forth in the Guide for the Care and Use of Laboratory Animals, National Research Council, 1996 (Institutional Animal Welfare Assurance #A-3125-01).

**Nanoplex formulation.** Peptide amphiphiles were synthesized[5] by N-carboxyanhydride ring-opening polymerization[54–56] of benzyl L-aspartate NCA and subsequent deprotection or methylation. Nanoplexes containing siRNA were prepared using a modified thin-film hydration[57] and ethanol injection[58] method. Peptide amphiphiles were dried under a stream of nitrogen and reconstituted with siRNA

under sonication (96:2:2 mol% cation:helper:PEG; N:P = 1). The resulting mixture was then diluted using 50 v/v% ethanol:water with sonication and subsequent polyplexes were purified by dialysis against ultrapure water and diluted in isotonic saline immediately prior to injection.

**DNA platination assay**. KP7B cells were seeded on 6 well plate and incubated for 24 h at 37 °C. These cells were then treated with cisplatin (25 μM), and subsequently incubated for 5 h at 37 °C. Afterward, fresh medium was added, followed by an additional 12 h of incubation at 37 °C. The medium was then removed and the cells were washed twice with PBS (1 mL), harvested by trypsinization (1 mL), and washed twice with 0.5 mL PBS. Solutions containing cells were centrifuged at 400×g for 5 min at 4 °C. The cell pellet was suspended in DNAzol (1 mL, genomic DNA isolation reagent, MRC). The DNA was precipitated with ethanol (0.5 mL), washed with 75% ethanol (0.75 mL × 3), and redissolved in 1 mL of 8 mM NaOH. The DNA concentration was determined by UV–vis spectroscopy and the platinum content was quantified by graphite furnace atomic absorption.

**Statistical analysis**. All $p$ values were calculated using a two-tailed student's $t$ test in Graphpad Prism unless otherwise specified. *, **, ***, and **** denotes $p \le$ 0.05, $p \le 0.01$, $p \le 0.001$, and $p \le 0.0001$, respectively. All error bars shown in Figures indicate standard error of the mean.

**Reporting summary**. Further information on research design is available in the Nature Research Reporting Summary linked to this article.

## Data availability

All data are available from the corresponding authors upon reasonable request. Source data are provided as a Source Data file. Source data are provided with this paper.

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

## Acknowledgements

We wish to thank Drs. Ian Cannell, Mun Kyung Hwang, Karl Merrick, Leny Gocheva, and all members of the Yaffe, Hammond and Hemann laboratories for helpful discussions. We thank the Robert A. Swanson (1969) Biotechnology Center, especially the Preclinical Modeling, Imaging & Testing Facility, the Flow Cytometry Facility, the Hope Babette Tang (1983) Histology Facility, Microscopy, and the Peterson (1957) Nanotechnology Materials Core Facility at the Koch Institute/MIT. This work was supported by grants from the National Institutes of Health (R01-ES015339, R35-ES028374, R01-CA226898, and R01-GM104047 to M.B.Y., NIBIB 1F32EB017614 to E.C.D., CA034992 to S.J.L. and O.H.Y., AG045144, CA211184 to O.H.Y.), the Ovarian Cancer Research Foundation (M.B.Y. and P.T.H.), the Breast Cancer Alliance (M.B.Y. and P.T.H.), US Department of Defense Congressionally Directed Medical Research Ovarian Cancer Research Program (P.T.H.; OCRP Teal Innovator Award; W81XWH-13-1-0151), the Charles and Marjorie Holloway Foundation (M.B.Y.), the STARR Cancer Consortium (M.B.Y. and M.T.H.), the Misrock Foundation (Y.W.K.), the MIT Center for Precision Cancer Medicine (M.B.Y., M.T.H., Y.W.K., and F.C.L.), and the Mazumdar-Shaw International Oncology Fellowship (G.S.). Support was provided in part by the Koch Institute Support Grant (P30-CA14051) from the National Cancer Institute, and the MIT MRSEC Shared Experimental Facilities Grant (DMR-0819762) from the National Science Foundation.

## Author contributions

Conceptualization: Y.W.K., E.C.D., P.T.H., and M.B.Y.; methodology: Y.W.K., E.C.D., P.T.H., and M.B.Y.; acquisition of data, Y.W.K., E.C.D., S.M., S.S.D., J.C.P., M.Q., A.D., F.C.L., G.S., K.E.S., S.V., H.C.R., and W.Z.; analysis and interpretation of data, Y.W.K., E.C.D., J.C.P., F.C.L., G.S., P.T.H., and M.B.Y.; writing, review, and/or revision of the manuscript, Y.W.K., E.C.D., O.H.Y., S.J.L., M.T.H., P.T.H., and M.B.Y.

## Competing interests

The authors declare no competing interests.
