## [Peer Review File · Nature Communications]

Reviewers' comments:

Reviewer #1 (Remarks to the Author):

The paper by Kong et al describes a type of synthetic lethality (SL) they term "augmented synthetic lethality" (ASL) in which the differential sensitivity of a p53 mutant lung adenocarcinoma cancer cell line to cis-platin is facilitated by the simultaneous co-targeting of MK2 and XPA. Previous work from the Yaffe laboratory had demonstrated a weaker but significant SL relationship between p53 and MK2 (a digenic negative interaction) in the context of DNA damaging chemotherapeutic agents. The current work purportedly extends these findings to a potent trigenic negative interaction between p53, MK2, and XPA by demonstrating that simultaneous loss of MK2 and XPA causes increased killing by cis-platin in p53-defective lung adenocarcinoma cells relative to targeting either pathway alone. The authors suggest this is due to the hyperactivation of MK2 signaling in XPA-deficient cells. They go on to utilize an RNAi delivery system developed in their laboratory, involving RNAi-peptide nanocomplexes, to simultaneously co-target MK2 and PKA in p53-defective tumors in vivo, and observe enhanced tumor control and improved long term survival in a fully immunocompetent mouse model of lung adenocarcinoma. The tumor cell killing effects of simultaneous targeting of MK2 and PKA and treating with cis-platin in this in vivo test system are dramatic.

The work clearly demonstrates a strong synthetic lethal relationship between MK2 and XPA in presence of cis-platin in the adenocarcinoma cell line. The authors claim that the lethality is dependent on the p53 status of the line, however they do not directly demonstrate that p53 mutation is necessary to observe the SL effect. The authors claim that the trigenic SL interaction between MK2, PKA, and p53, that they call ASL, is novel, and stress that p53 mutations are prevalent in many cancer types suggesting broad applicability for differential killing relative to p53-WT cells. Trigenic negative genetic interactions are not a novel concept, so I am not convinced that a conditional trigenic interaction (causing drug sensitivity to a sub-lethal dose of a DNA damaging agent) warrants creating a new nomenclature term (ASL), although this is not a major concern. Nevertheless, the authors do not formally demonstrate the requirement of p53 loss of function in the MK2/PKA co targeting experiments, so the ASL concept has not been demonstrated. The claim that the genetic context of p53-mutant vs p53-WT is critical to demonstrating ASL, but it is not tested. Would introduction of p53 back into KP7B or H1299 cancer cell lines rescue the lethality of simultaneous knockdown of XPA and MK2?

The second half of the paper uses RNA-peptide nanoplexes to downregulate genes in vivo. The authors convincingly show that a combination of siRNAs targeting XPA and MK2 shrinks p53-deficient xenografts in the presence of cisplatin. However, the effect in the absence of cisplatin or in p53-proficient tumors is not investigated. It would also be beneficial to know how the treatment might affect normal mice (by eg measuring their weight gain/loss over treatment or other phenotypes) as the treatment involves a combination of knockdown of two genes and a potent genotoxic agent. The authors claim that:

MK2 has a well-established role in innate immunity and inflammatory signaling (26), similar to DNA-PK (35), raising the possibility of off-target side effects from systemic inhibition. To overcome these limitations, we developed a novel tumor-targeting nanoparticle that delivers siRNAs to the site of pre-existing MK2-containing tumors and efficiently depletes both its target RNA and protein. (page 12).

It is not easy to determine from the text why nanoparticles would target tumors specifically and not also affect immune cells and no data for this appears to be given. If the authors are using nanoparticles to avoid side effects by specifically targeting tumor cells, this should be clarified.

Additional points:

-Many of the methods seem incomplete. Eg for survival assays, the authors write: Cell viability was measured using CellTiter-Glo luminescent cell viability assay as per manufacturer's instructions (Promega) (pg16)

It would also be useful to know how long the cells were grown and when/how siRNAs and drugs were added.

Materials and methods lack explanations about how "expected" values are calculated.

-Figure 1- the authors switch between calling pMK2 levels a measure of MK2 activity and/or MK2 activation which is confusing

-Figure 1D- the pMK2 protein levels should be normalized to the levels of MK2 in the blot; it's not clear if this was done

-Figure 2A- the symbols in the graph legend do not match the ones in the figure

-Figure 2C- the DAPI staining in the nuclei appears abnormally punctate bringing into question the viability of these cells; additionally, gamma-H2AX foci were counted for Figure 2D and it is unclear how a cell like the one at the bottom of the siMK2/XPA picture could even have their foci individualized to be counted. H2AX foci numbers do not correlate well with the quantification graph in Fig 2D.

-Figure 2D- for siM/X, please write out in full (also Figure 4D)

-Although the authors show dual knockdown by western blot in Figure 4, it might also be nice to see that dual knockdown occurs at a single cell level (using eg immunofluorescence)

-Figure 4F-the authors claim in the text that the dual siRNA-containing nanoplexes are "more than doubling the survival of longest lived animals in any of the other cohorts" (pg 11), and perhaps the x-axis should be changed to "Time (post-treatment)" to better reflect what I think this statement means

-Supplemental Figure 1D and E- it is not explained where the expected line on the graph came from and it seems like data is missing (eg where are the data for the effect of each single siRNA alone); should cisplatin be relabeled siControl?

-Supplemental Figure 2C- could use a better explanation for what is increasing, maybe the concentration of the nanoplexes?

- Sup fig 1B & C are not mentioned in the text. Why the DOX experiments

Reviewer #2 (Remarks to the Author):

In this study, Kong et al., expand on data indicating that loss of MK2 promotes sensitivity to DNA damaging agents only in the absence of p53. Augmented synthetic lethality shows that depletion of a third gene, XPA, enhances the synthetic lethality.

Specific comments:

1. Figures S1B and C are not cited in the text.
2. Text page 10 says "allow the tumors to grow larger and the cancer even to be more advanced..." What are the data that indicate the larger tumors are more advanced?

3. The MK2 and pMK2 blots differ in figures 1B and C (one, two or three bands are shown). Please explain these differences. Also in figure 1B, is the upper MK2 band representative of active MK2? The description is not clear.
4. Also, the western blots in figures 1B and C should have actin controls for each blot that was probed unless all of the westerns represent a single blot.
5. While the experiments are well done with appropriate controls, only one lung cancer cell line, KP7B was used. As thus, the generality of this augmented synthetic relationship is limited.
6. In figure 2A, the symbols do not match. For example, the red line has triangles but triangles are blue and the red line has squares. Thus, I am not sure which line represents which experiment.
7. In figure 2D, it appears that there might be statistical differences that were not noted. Also please indicate what columns are being compared for the asterisks indicated.
8. It is surprising that in this synthetic lethal interaction, not all cells die. Do nanoplexes target all cells? If not, what percent of cells are targeted?

Reviewer #3 (Remarks to the Author):

The authors have developed a new approach for improving synthetic lethality induction in NSCLC using polymeric siRNA nanocomplexes. The study design is systematic and nicely articulated. However, there are several concerns with experimental controls and analysis as mentioned below. These needs to be addressed:

1. In line 264 the model is described in Figure 3D and not 3A as written in the text.
2. The authors should conduct a more precise quantification of in vivo siRNA delivery to the tumors via siRNA specific qPCR or any other quantitative method (but not fluorescently based method) in order to really know how much of the siRNAs are reaching the tumors and how much is really functional.
3. The authors should use a MK2 small molecule inhibitor as a control in all in vivo experiments to demonstrate the superiority of the siRNA. Furthermore, the toxicity of this control should also be evaluated.
4. The authors should evaluate the in vivo silencing of MK2 and XPA by the nanoplexes as they conducted in Figure 3F-H. Furthermore, the authors should further validate the mechanism of the combination of siMK2 and siXPA to induce synthetic lethality in vivo as they have conducted in Figure 3S.

We are very grateful to the reviewers for their careful consideration of our manuscript. In response to their critique, we have now addressed all of the comments, leading to substantial textual revisions, and inclusion of 14 new figure panels, which we believe have significantly improved the manuscript. A point-by-point response to each item raised in the critique follows below.

Reviewers' comments:

Reviewer #1:

(1) The claim that the genetic context of p53-mutant vs p53-WT is critical to demonstrating ASL, but it is not tested. Would introduction of p53 back into KP7B or H1299 cancer cell lines rescue the lethality of simultaneous knockdown of XPA and MK2?

This is an interesting suggestion. Unfortunately, simply re-introducing functional p53 back into p53-deficient tumor cell lines results in rapid senescence or apoptosis of the cells *in vitro* (c.f. Bykov et al., *Nature Medicine* 2002 8:282-288), and *in vivo* (Martins et al., 2006 *Cell* 127:1323-1334). This effect has been well demonstrated in p53 lox-stop-lox and K-Ras G12D lox-stop-lox tumors, (Ventura et al. *Nature*, 2007 445:661-665), including NSCLC, particularly in the setting of DNA damage (Juntilla et al., *Nature* 2010 468:567-571). Therefore, it is technically not possible to perform the direct experiment suggested by the reviewer. Nonetheless, to try and address the reviewer's important point about whether the genetic context of p53 is important for ASL, we have now tested our siMK2/siXPA combination in NSCLC cell lines bearing either a p53 wild-type genotype (H1563; Fig. S2C), a p53 mutant genotype (H2009; Fig. S2A) or a p53 Null genotype (H1299; Fig. S2B), as well as in the pair of isogenic HCT116 p53-WT and HCT116 p53-null colon cancer cell lines (see Fig below). As shown in Figure S2, the combined knock-down of XPA and MK2 results in significantly less cell survival after cisplatin treatment in both the p53 mutant and null NSCLC cell lines, but not in the p53 wild-type cell line (compare the blue, red, and purple bars). In fact, in the p53 wild-type cell line, knocking down both XPA and MK2 was less effective than just knocking-down MK2 alone. These findings are described on lines 184-188 of the revised manuscript.

Similar results are shown below for the isogenic p53 wild-type and null colon cancer cell line HCT116, where again the combined knock-down of XPA and MK2 markedly reduces cell viability after cisplatin treatment, beyond that seen with individual knockdown of XPA or MK2 alone, only in the p53-null colon cancer cell line. (The dotted lines show the expected cell viability calculated based on the Bliss independent model of additivity: siMK2 viability X siXPA viability X cisplatin viability, following Foucquier and Guedj, 2015, *Pharmacol Res Perspect* 3, e00149.)

These new data on XPA and MK2 are in strong agreement with our prior work showing that, in the setting of DNA damage by cisplatin or doxorubicin, MK2 loss itself is synthetic lethal with loss of functional p53 (c.f. Reinhardt et al., *Cancer Cell* 2007, 11:175-189; Cannell et al., *Cancer Cell* 2015 28:623-637).

(2) The second half of the paper uses RNA-peptide nanoplexes to downregulate genes in vivo. The authors convincingly show that a combination of siRNAs targeting XPA and MK2 shrinks p53-deficient xenografts in the presence of cisplatin. However, the effect in the absence of cisplatin or in p53-proficient tumors is not investigated.

This was a great idea! We have historically been very focused on improving the therapeutic use of conventional chemotherapy (i.e. cisplatin, doxorubicin) in cancer, and had not thought to explore the effect of combined XPA and MK2 knock-down alone in our *in vivo* NSCLC model. We have performed the requested experiments, and now present the results in the new Figure S8C. Remarkably, the combined XPA and MK2 knockdown itself resulted in a significant extension of survival even in the absence of cisplatin treatment, although the addition of cisplatin further significantly increased the median survival by an additional 15 days. These findings are entirely consistent with the XP cell line data shown in Figures 1B and S1A, where loss of XPA, and other NER genes hyperactivates MK2 even in the absence of cisplatin treatment, though the hyperactivation is significantly further enhanced by cisplatin exposure. These findings are now described on lines 331-336 of the revised manuscript. Thank you for suggesting this excellent experiment!

(3) It would also be beneficial to know how the treatment might affect normal mice (by eg measuring their weight gain/loss over treatment or other phenotypes) as the treatment involves a combination of knockdown of two genes and a potent genotoxic agent.

Thank you for this suggestion. As requested, we have measured the weights of tumor and non-tumor bearing mice that were treated with our combination. These new data are presented in Fig. S7 B and C. Tumor bearing mice were given either siControl, siMK2, siXPA, or siMK2+siXPA in combination with cisplatin. As shown in panel C, tumor-bearing mice that received the combination siMK2+siXPA+cisplatin treatment had the least amount of weight loss amongst the 4 treatment groups. In panel B, non-tumor-bearing mice were given saline (mock), cisplatin alone, or cisplatin in combination of MK2 and XPA siRNA. Mice that were given saline had an overall weight gain of ~8% while the mice that received cisplatin had an overall weight loss of ~5% (Fig. S7B). Mice that received the combination siMK2+siXPA+cisplatin treatment maintained their pre-treatment weight. These results are described in lines 302-305 in the revised manuscript.

(4) The authors claim that:

MK2 has a well-established role in innate immunity and inflammatory signaling (26), similar to DNA-PK (35), raising the possibility of off-target side effects from systemic inhibition. To overcome these limitations, we developed a novel tumor-targeting nanoparticle that delivers siRNAs to the site of pre-existing MK2-containing tumors and efficiently depletes both its target RNA and protein. (page 12).

It is not easy to determine from the text why nanoparticles would target tumors specifically and not also affect immune cells and no data for this appears to be given. If the authors are using nanoparticles to avoid side effects by specifically targeting tumor cells, this should be clarified.

To experimentally address this point, and directly examine whether the nanoparticles were taken up preferentially by tumor cells, the immune cells, or both, we co-cultured 500,000 splenocytes with 500,000 KP7B tumor cells and then treated the cultures with Alexa Fluor 647 (AF647)-fluorescently tagged-siRNA-nanoplexes for 24h (Fig S3G) (c.f. Nyland, J. F., Bai, J. J. K., Katz, H. E. & Silbergeld, E. K. *In vitro* interactions between splenocytes and dansylamide dye-embedded nanoparticles detected by flow cytometry. *Nanomed-Nanotechnol* **5**, 298-304, doi:10.1016/j.nano.2009.01.001 (2009)). Viable cells were then analyzed by flow cytometry for uptake of the fluorescent siRNAs, along with

identification of the specific cell populations by co-staining with fluorophore conjugated antibodies against CD45, CD11b, F480 and CD3. CD45+CD11b+AF647+ cells were scored as siRNA+ macrophages, CD45+CD3+AF647+ cells were scored as siRNA+ T-cells and CD45-AF647+ cells were scored as siRNA+ tumor cells. This new data is now shown in Figure S3G, and described in lines 224-227 and 899-904 in the revised text. We found that the nanoplexes showed essentially no uptake in T-cells, and only very modest uptake in macrophages, compared to their uptake in the tumor cells. The molecular basis for this is not clear, but probably results from the large amount of macropinocytosis used by K-Ras-driven tumor cells to facilitate nutrient scavenging (Recouvreur and Comisso, *Front. Endocrinol.* 2017 8:261), in contrast to non-transformed cell types. *In vivo*, this tumor cell-enriched uptake is likely to be further exploited by the preferential targeting of the nanoparticles to tumors as a consequence of so-called ‘passive’ targeting (c.f. Harrington, K. J. *et al.* Effective targeting of solid tumors in patients with locally advanced cancers by radiolabeled pegylated liposomes. *Clin Cancer Res* 7, 243-254 (2001)). Nanoparticles are able to exploit the cancer’s distinct vascular and lymphatic pathology (i.e. leaky vasculature and defective lymphatic drainage) to accumulate in the tumor. Here we utilize the enhanced tumor cell uptake and passive tumor targeting properties to formulate nanoparticles that are able to pass through the leaky vascular junctions in tumors resulting in preferential accumulation at the tumor site over time, as shown in Figure S4.

Additional points:

1) *Many of the methods seem incomplete. Eg for survival assays, the authors write: Cell viability was measured using CellTiter-Glo luminescent cell viability assay as per manufacturer’s instructions (Promega) (pg16). It would also be useful to know how long the cells were grown and when/how siRNAs and drugs were added.*

Thank you for pointing this out. We have now added all of the missing methods to the text and expanded the entire Methods section. For cell viability assays, cells were transfected using RNAiMax (Life Technologies) for 48 hours prior to treatment with cisplatin for 72h. Cell viability was measured using CellTiter-Glo assay. This is now described explicitly in lines 466-470 of the revised text.

2) *Materials and methods lack explanations about how “expected” values are calculated.*

As now described in lines 470-471 of the revised text, the “expected” values were calculated based on the bliss independence model of additivity. (siRNA1 viability X siRNA2 viability X Cisplatin viability), using the method outlined by Fouquier and Guedj, 2015, *Pharmacol Res Perspect* 3, e00149, which has also been added as a reference (reference 48).

3) *Figure 1- the authors switch between calling pMK2 levels a measure of MK2 activity and/or MK2 activation which is confusing.*

Thanks for pointing this out. We have now changed “MK2 activity” to “MK2 activation” throughout the text to avoid confusion, particularly since the phosphorylation site corresponds to MK2 activation.

4) *Figure 1D- the pMK2 protein levels should be normalized to the levels of MK2 in the blot; it’s not clear if this was done.*

In the prior submission, the pMK2 protein levels were normalized to vinculin. We have now repeated the analysis by normalizing the pMK2 protein level to total MK2, as requested. These data are presented in the new Fig. 1E, and the Y-axis labelled to indicate this normalization.

5) *Figure 2A- the symbols in the graph legend do not match the ones in the figure.*

Thank you for picking this up! We have now changed the symbols in the graph legend to match the ones shown in the figure.

- 6) *Figure 2C- the DAPI staining in the nuclei appears abnormally punctate bringing into question the viability of these cells.*

Thank you for your comment. Below are representative images of KP7B cells that have **not** been treated with cisplatin, but have been stained with DAPI. We find that these cells, which are fully viable, have similar pattern of euchromatin and heterochromatin as that present in the treated cells shown in Figure 2C. KP7B cells were created by, and provided to us by the lab of Tyler Jacks, located two floors above us, thus their nuclear morphology is unlikely to reflect an artifact. Instead, we hypothesize that the DAPI staining pattern is likely due to increased heterochromatin in this cell line.

- 7) *additionally, gamma-H2AX foci were counted for Figure 2D and it is unclear how a cell like the one at the bottom of the siMK2/XPA picture could even have their foci individualized to be counted. H2AX foci numbers do not correlate well with the quantification graph in Fig 2D.*

We apologize for any confusion. The prior graph shown in the original Figure 2D was from an experiment performed under the identical cisplatin treatment conditions as those that were used to analyze for repair of platinum-DNA adducts in Figure 2B. In those experiments, cell were treated with cisplatin for 5 hrs, the drug washed off, and the cells analyzed 12 hours after the wash out using a high-throughout automated microscope and quantified using Cell Profiler, as described previously (Floyd et al., Nature 2013 498:246-250) and in lines 510-518 of the revised manuscript. The example micrograph that we showed in the former Figure 2C was taken, instead, using a Deltavision deconvolution microscope from a similar experiment to better photograph the foci, and make the point that combined knockdown of XPA and MK2 results in increased γ H2AX foci, except that in this experiment the cisplatin was not washed off, but remained on the cells for 24 hours. We should have made this difference between panels C and D clearer in the Figure captions. In order to avoid any confusion, however, we have now used the same Cell Profiler analysis to quantify the Deltavision deconvolution images from multiple experiments performed using the same conditions as those shown in the micrograph in Figure 2C, with continuous exposure to cisplatin or 24 hours. All conditions are now better described in the Figure 2 caption (lines 823-827), and the methods section (lines 510-518) of the revised manuscript.

- 8) *Figure 2D- for siM/X, please write out in full (also Figure 4D)*

This has been corrected to siMK2+siXPA.

9) *Although the authors show dual knockdown by western blot in Figure 4, it might also be nice to see that dual knockdown occurs at a single cell level (using eg immunofluorescence)*

Thank you for your suggestion. As requested by the reviewer, we performed immunofluorescence to verify dual knockdown of MK2 and XPA at a single cell level and have added these data to Fig. S3D-F. Fig. S2D show representative images of KP7B cells treated with nanoplex-siMK2, nanoplex-siXPA or nanoplex-siMK2/XPA and stained with MK2 and XPA antibodies. KP7B cells treated with nanoplex-siMK2/XPA showed loss of both MK2 and XPA staining at a single cell level (Figure S3D). The integrated intensity of MK2 and XPA staining in individual KP7B cells treated with nanoplex-siMK2, nanoplex-siXPA, and nanoplex-siMK2/XPA is quantified in Figure S3E, and the percentage of cells staining positively for MK2 and XPA after single and combined knock down is shown in Figure S3F. This is now described in lines 220-227, 518-519, and 895-899 of the revised manuscript. We find that >95% of cells treated with the nanoplex-siMK2/XPA combination *in vitro* show dual knock down of both MK2 and XPA.

10) *Figure 4F-the authors claim in the text that the dual siRNA-containing nanoplexes are “more than doubling the survival of longest lived animals in any of the other cohorts” (pg 11), and perhaps the x-axis should be changed to “Time (post-treatment)” to better reflect what I think this statement means*

Thank you for your suggestion. With all due respect, we believe that changing the x-axis to “Time (post-treatment)” in Fig. 4F will result in the confusion with the other figures in the paper. We have, however, added a timeline for this dual siRNA-containing nanoplex experiment in the new Figure S7A, similar to the timeline shown in Figure 3E, to provide additional clarity.

11) *Supplemental Figure 1D and E- it is not explained where the expected line on the graph came from and it seems like data is missing (eg where are the data for the effect of each single siRNA alone); should cisplatin be relabeled siControl?*

As explained above in point 1, we have now replaced these panels with Supplemental Figure 2A-D, where we better illustrate the enhanced effect of combined MK2 and XPA knock-down in p53 null or mutant NSCLC cell lines, compared to a p53 wild-type NSCLC cell line, and have included the data with the siControls, and the individual knock downs.

12) *Supplemental Figure 2C- could use a better explanation for what is increasing, maybe the concentration of the nanoplexes?*

Thank you for your comment. In that Figure (now Fig. S3C), we are increasing the N:P ratio, which is the ratio of positively-chargeable polymer amine (N=nitrogen) groups to negatively-charged nucleic acid (siRNA) phosphate (P) groups, thereby increasing the nanoplexes:siRNA ratio. We have included a brief description in methods section, under cell toxicity studies (lines 522-524), and explained this in the revised caption to Figure S3C (lines 894-895).

13) *Sup fig 1B & C are not mentioned in the text. Why the DOX experiments?*

Thank you for catching this oversight! We have had a long-standing interest in the role of MK2 signaling in DNA damage responses to both cisplatin and doxorubicin. Interestingly, we see a similar increase of phospho-MK2 in XPA deficient cells when cells are treated with doxorubicin. Since the doxorubicin effect is not within the scope of this paper, we have now removed that data (previously Fig. S1B). The previous Fig. S1C showed the efficiency of knockdown of MK2 and XPA in KP7B cells with lipofectamine RNAiMax for comparison with the extent of knockdown that we obtain with

the nanoplex constructs. We have now moved that panel back to Fig. 4B, to directly compare the efficiency of dual MK2/XPA knockdown by the nanoplexes vs Lipofectamine RNAiMax.

Reviewer #2:

1. *Figures S1B and C are not cited in the text.*

Thank you for catching this oversight! As we noted in our response to Reviewer 1, point 13, we have had a long-standing interest in the role of MK2 signaling in DNA damage responses to both cisplatin and doxorubicin. Interestingly, we see a similar increase of phospho-MK2 in XPA deficient cells when cells are treated with doxorubicin. Since the doxorubicin effect is not within the scope of this paper, we have now removed that data (previously Fig. S1B). The previous Fig. S1C showed the efficiency of knockdown of MK2 and XPA in KP7B cells with lipofectamine RNAiMax for comparison with the extent of knockdown that we obtain with the nanoplex constructs, since the panels were from the same blot. We have now moved that panel back to Fig. 4B to directly compare the efficiency of dual MK2/XPA knockdown by the nanoplexes vs Lipofectamine RNAiMax.

2. *Text page 10 says “allow the tumors to grow larger and the cancer even to be more advanced...” What are the data that indicate the larger tumors are more advanced?*

This is a very good point. We allowed the tumors to develop for 3 weeks compared to the 2 weeks in the earlier experiments. The median survival for the control mice post-first treatment initiated after 3 weeks is 17 days (Figure 4F, black line) compared to 32 days if treatment begins after 2 weeks (Figure 3N, black line) suggesting that there is an increased tumor burden when we begin the treatment after 3 weeks. We did not, however, grade/stage the tumors prior to initial treatment, so we cannot say with certainty that the tumors were more advanced. We have therefore removed the text “and the cancer to be more advanced”, as suggested by the reviewer.

3. *The MK2 and pMK2 blots differ in figures 1B and C (one, two or three bands are shown). Please explain these differences. Also in figure 1B, is the upper MK2 band representative of active MK2? The description is not clear.*

We apologize for any confusion. MK2 is ubiquitously expressed (Gaestel, 2006, reference 34) and a multiple band pattern is usually detected when analyzed by western blotting (see below). Former Figure 1B (now 1B and C) shows human cells which display 2 prominent bands. Former Figure 1C (now Figure 1D), are mouse cells and similarly show 2 prominent bands (as well as a faint minor band that may be a breakdown product of the larger band). The origin of these two bands was very recently elucidated by Trulley et al. (Trulley, P. *et al.* Alternative Translation Initiation Generates a Functionally Distinct Isoform of the Stress-Activated Protein Kinase MK2. *Cell Rep* 2019, **27**:2859, doi:10.1016/j.celrep.2019.05.024). There are two alternative translation initiation start codons, a CUG start codon located 141 nt upstream of the canonical AUG initiation codon, which gives rise proteins of 386 or 433 amino acids that account for the two bands seen on SDS-PAGE. The ratio of expression of these bands varies between tissues and cell lines (Trulley et al., 2019, see below), and the mechanisms responsible for this differential regulation have not yet been worked out. (There is even a third upstream potential GUG start codon located 309nt upstream of the canonical AUG initiation codon, although it appears to be rarely used). Similarly, functional differences between the two isoforms are not very well understood. The differences observed in our paper are likely due to the different banding patterns between mouse (KP7B) and human cells (human fibroblast) as well as the tissue origin of the cells. Importantly, in MK2 knockout MEFs (A), both bands (lane 2 vs lane 8)

disappear, indicating that both bands are MK2. We showed similar loss of both bands in our previously published MK2 knock-out animals (Figures 2 and 4 in Morandell S. et al., A Reversible Gene-Targeting Strategy Identifies Synthetic Lethal Interactions between MK2 and p53 in the DNA Damage Response In Vivo. *Cell Reports* 2013 5:868-77). We now comment in the revised caption to Figure 1 that the multiple banding pattern arises from alternative translation start sites (lines 810-811) and cite the Tulley et al., paper (reference 54).

Figure taken from Trulley et al., 2019.

4. Also, the western blots in figures 1B and C should have actin controls for each blot that was probed unless all of the westerns represent a single blot.

As requested, we have now added the actin or vinculin blots for Fig. 1B (now Figure 1C) in Fig. S1G. For Figure 1C (now Figure 1D) we ran the same samples over several gels and performed Ponceau-S staining to ensure equal loading.

5. While the experiments are well done with appropriate controls, only one lung cancer cell line, KP7B was used. As thus, the generality of this augmented synthetic relationship is limited.

Thank you for pointing this out. Reviewer 1 made a related point (R1 point 1). We have therefore now expanded our analysis to include a number of additional NSCLC cell lines. As now shown in the new Supplemental Figure S2, we tested our siMK2/siXPA combination in NSCLC p53 wild-type (H1563), p53 mutant (H2009) and p53 Null (H1299) lung cancer cells (Fig S2A-C). These new data show that the combined knock-down of XPA and MK2 results in significantly less cell survival after cisplatin treatment in both the p53 mutant and null NSCLC cell lines, but not in the p53 wild-type cell line (compare the blue, red, and purple bars). In fact, in the p53 wild-type cell line, knocking down both XPA and MK2 was less effective than just knocking-down MK2 alone. These findings are described on lines 184-188 of the revised manuscript

We also examined single and combined MK2 and XPA knock-down in the isogenic HCT116 p53 WT and HCT116 p53 Null colon cancer cells (shown below). Again, the combined knock-down of XPA and MK2 markedly reduces cell viability after cisplatin treatment, beyond that seen with individual knockdown of XPA or MK2 alone, only in the p53-null colon cancer cell line. (The dotted lines show the expected cell viability calculated based on the Bliss independent model of additivity: siMK2 viability X siXPA viability X cisplatin viability, following Fouquier and Guedj, 2015, *Pharmacol Res Perspect* 3, e00149.), indicating that siMK2 and siXPA synergizes with cisplatin in p53 mutant/null cells but not with p53 WT cells.

6. In figure 2A, the symbols do not match. For example, the red line has triangles but triangles are blue and the red line has squares. Thus, I am not sure which line represents which experiment.

We apologized for the mislabeling. We have now corrected it.

7. In figure 2D, it appears that there might be statistical differences that were not noted. Also please indicate what columns are being compared for the asterisks indicated.

We have replaced the figure with a graph that more accurately represents the images in Fig. 2C and added the p-value. (See also comments to reviewer 1, point 7). The p-values are now indicated in the figure legend (lines 826-827 in the revised text).

8. It is surprising that in this synthetic lethal interaction, not all cells die. Do nanoplexes target all cells? If not, what percent of cells are targeted?

The lung adenocarcinoma mouse model that we are using is extremely aggressive. Untreated mice generally die within 5 weeks of tumor implantation of 20,000 cells. We believe that not all tumor cells die because not all tumor cells take up the nanoplexes. From the data obtained from our nanoplexes-fluorescently tagged siRNA studies (Fig. S4A-C), we know that approximately 5% of the initial siRNA injected into mice is taken up by the lung tumors, and from *in vitro* assays performed to determine uptake of nanoplexes by tumor cells vs immune cells (Fig. S3G), we know that up to 20% of the tumor cells show significant nanoplex uptake after 24 hours under these *in vitro* conditions. Furthermore, our immunofluorescence studies (new Figure S3D-F) show that the majority of cells that take up the nanoplexes will show dual knockdown. This suggests that if we can improve nanoplex uptake by the tumor cells, we can achieve better tumor killing. We have added two sentences to this effect in the revised discussion (lines 380-383).

Reviewer #3:

The authors have developed a new approach for improving synthetic lethality induction in NSCLC using polymeric siRNA nanocomplexes. The study design is systematic and nicely articulated. However, there are several concerns with experimental controls and analysis as mentioned below. These need to be addressed.

1. In line 264 the model is described in Figure 3D and not 3A as written in the text.

We apologize for the error. We have now corrected this in the text (line 296) of the revised text.

2. The authors should conduct a more precise quantification of in vivo siRNA delivery to the tumors via siRNA specific qPCR or any other quantitative method (but not fluorescently based method) in order to really know how much of the siRNAs are reaching the tumors and how much is really functional.

Thank you for this suggestion. This is technically not possible to do without destroying the tumor in the process. However, to determine if the siRNA delivered to the tumors are functional, we performed western blot analysis on the residual tumors at the time the animals were sacrificed to show that MK2 and XPA proteins were knocked down in siRNA nanoplex-treated the tumors (Figure 3G and S8B).

3. The authors should use a MK2 small molecule inhibitor as a control in all in vivo experiments to demonstrate the superiority of the siRNA. Furthermore, the toxicity of this control should also be evaluated.

Thank you for this suggestion. There are currently no commercially available inhibitors that specifically inhibit only MK2 (all MK2 inhibitors available to date also target MK3 and MK5 as well). Nonetheless, to try and address the reviewer's point, we performed experiments using an MK2 inhibitor from Hubbard Biomedical as a control. In our hands, we found that their inhibitor alone was quote toxic at relevant concentrations, and the mice died before its therapeutic efficiency could be evaluated, possibly as a consequence of co-inhibiting MK3 and MK5. In contrast, MK2 whole body and conditional knock-out animals are fully viable (Gaestel, 2006, reference 34, and Suarez-Lopez et al., *Proc Natl Acad Sci USA*. 2018 115:E4236-E4244).

4. The authors should evaluate the in vivo silencing of MK2 and XPA by the nanoplexes as they conducted in Figure 3F-H. Furthermore, the authors should further validate the mechanism of the combination of siMK2 and siXPA to induce synthetic lethality in vivo as they have conducted in Figure 3S.

As suggested we have now directly examined silencing of both MK2 and XPA by nanoplexes, using Western blot analysis of both cells and in residual tumors from animals co-targeted by siMK2/siXPA nanoplexes at the conclusion of the in vivo animal experiments (Figures 4B and S8B). We also used immunofluorescence to demonstrate co-knockdown of both MK2 and XPA at the level of individual cells (new Figure S3D-F). (See also response to Reviewer 1 point 9). We also further validated the mechanism of the combination of siMK2 and siXPA to induce synthetic lethality specifically in p53-defective cells by using a second set of siRNA in KP7B cells, as well comparing H1299 (p53 null), H2009 (p53 mutant), and H1563 (p53 wild-type) cells, as shown in Supplementary Figure S2. In addition, we also validated this dependency using isogenic p53 wild-type or null HCT 116 cells, as shown above in response to Reviewer 1 point 1, and Reviewer 2 point 5.

REVIEWERS' COMMENTS:

Reviewer #1 (Remarks to the Author):

The paper by Kong et al has undergone substantial revision, including new experiments to address my, and the other reviewer's, comments. Overall, the paper is substantially improved. The authors should modify the text (lines 184-188), in describing the strength/synergism of the trigenic interaction between MK2, PKA, and p53 in the presence of cis-platin, that they have called ASL.

The work demonstrates a synthetic lethal relationship between MK2 and XPA in presence of cis-platin in the adenocarcinoma cell line (Fig2A). The authors have addressed my concern about their claim that the lethality is dependent on the p53 status of the line by testing for synthetic sensitivity to cis-platin in three p53-minus cell lines as compared to one p53+ cell line (Fig S2). This is supportive of their model, but not conclusive, given that the genetic backgrounds of the four cell lines certainly differ in many ways in addition to their p53 status. A better experiment is presented in the rebuttal document (comparing results obtained in a matched cell line pair, ie., HCT116 p53+ vs HCT116 p53-). This experiment should be included in the paper (e.g. as Fig S2 E,F). In this case the p53 null line shows significantly reduced growth for the double KD, as compared to either single KD. The effect appears slightly greater than additive. One thing to note: the "expected" growth of the double KD in the p53 null HCT116 in 25µm Cisplatin appears too high relative to the single KD's (dotted line).

I would suggest modifying the sentence (lines 183-188) by deleting "similar synergistic killing" and as follows:

Additive or greater than additive killing by cisplatin after combined MK2 and XPA knockdown, compared to individual knockdowns, was also observed in three p53 negative human NSLC tumor cell lines (H2009, H1299, K97B), but not observed in a p53+ NSCLC cell line.

I would also suggest adding the HCT116 matched pair cell line results here.

As noted previously, trigenic negative genetic interactions are not a novel concept, so I am not convinced that a conditional trigenic interaction (causing drug sensitivity to a sub-lethal dose of a DNA damaging agent) warrants creating a new nomenclature term (ASL), although this is not a major concern. The authors should at least reference somewhere the trigenic genetic interaction paper published recently in Science from the Boone/Andrews labs (PMID: 29674565).

Reviewer #3 (Remarks to the Author):

The authors have address my major concerns.

Reviewer #4 (Remarks to the Author):

In the revised manuscript and accompanying rebuttal letter, the authors have adequately responded and in a satisfactory manner to all major concerns and questions of this reviewer and to the comments of the reviewers as well as fa as I can judge.

With these changes and additions to the manuscript has certainly gained in quality and impact.

I do have a few additional remarks, not raised before by the other reviewers, concerning mainly the DNA repair aspects.

1) On page 5, lines 143-144; the authors state that: '...resulting in the cancer-prone syndromes XP and CS.', which is not correct and should be corrected. Indeed, CS is caused by a defect in a sub pathway of NER (TC-NER), but CS patients are not cancer-prone (PMID: 1308368; PMID:

19809470; PMID: 23428416; PMID: 26204423).

2) lines 181-182, the authors state: "Cells depleted of either MK2 or XPA alone showed modestly increased sensitivity to cisplatin as expected...". I do not understand this statement, as it is opposite to expectation, when XPA is depleted (which interferes with the major pathway to remove the major Cis-Pt DNA adduct (intra-strand crosslinks), as the authors state on lines 138-14300 it is expected that cells are severely sensitized towards this agent. Similarly, it is surprising to note in 2B that after depletion of XPA removal of Cis-Pt adducts is hardly affected, which again does not match with their statement that NER: 'Cells genetically lacking key NER proteins are unable to repair Pt-induced lesions...' (line 142). The authors should explain.

3) The rebuttal to reviewer #1, point 4 is important to include in the revised manuscript, as it is indeed enigmatic to the reader (including this reviewer) why the nanoparticles would specifically target cancer cells.

Point to point response to reviewer's comments.

Reviewer #1: The paper by Kong et al has undergone substantial revision, including new experiments to address my, and the other reviewer's, comments. Overall, the paper is substantially improved. The authors should modify the text (lines 184-188), in describing the strength/synergism of the trigenic interaction between MK2, PKA, and p53 in the presence of cis-platin, that they have called ASL.

The work demonstrates a synthetic lethal relationship between MK2 and XPA in presence of cis-platin in the adenocarcinoma cell line (Fig2A). The authors have addressed my concern about their claim that the lethality is dependent on the p53 status of the line by testing for synthetic sensitivity to cis-platin in three p53-minus cell lines as compared to one p53+ cell line (Fig S2). This is supportive of their model, but not conclusive, given that the genetic backgrounds of the four cell lines certainly differ in many ways in addition to their p53 status. A better experiment is presented in the rebuttal document (comparing results obtained in a matched cell line pair, ie., HCT116 p53+ vs HCT116 p53-). This experiment should be included in the paper (e.g. as Fig S2 E,F). In this case the p53 null line shows significantly reduced growth for the double KD, as compared to either single KD. The effect appears slightly greater than additive. One thing to note: the "expected" growth of the double KD in the p53 null HCT116 in 25µm Cisplatin appears too high relative to the single KD's (dotted line).

Thank you for this suggestion. As requested, we have now added the data from the HCT116 p53 WT and p53 null cells that was in the rebuttal letter into the manuscript as Supplementary Figure 2E and F, and described the results in lines 236-240 of the revised text.

I would suggest modifying the sentence (lines 183-188) by deleting "similar synergistic killing" and as follows: Additive or greater than additive killing by cisplatin after combined MK2 and XPA knockdown, compared to individual knockdowns, was also observed in three p53 negative human NSLC tumor cell lines (H2009, H1299, K97B), but not observed in a p53+ NSCLC cell line.

As requested, we have now modified the sentence as suggested. The new text on lines 231-234 of the revised text reads as follows: "*Additive or greater than additive killing by cisplatin after combined MK2 and XPA knockdown, compared to individual knockdowns, was also observed in three p53 deficient NSCLC tumor cell lines (H2009, H1299, and KP7B), but not observed in a p53+ cell line (H1563) (Fig. S2A-C).*"

I would also suggest adding the HCT116 matched pair cell line results here.

We have followed the reviewer's suggestion and included the relevant text in lines 236-240 as noted above.

As noted previously, trigenic negative genetic interactions are not a novel concept, so I am not convinced that a conditional trigenic interaction (causing drug sensitivity to a sub-lethal dose of a DNA damaging agent) warrants creating a new nomenclature term (ASL), although this is not a major concern. The authors should at least reference somewhere the trigenic genetic interaction paper published recently in Science from the Boone/Andrews labs (PMID: 29674565).

We believe that the trigenic interaction between p53, MK2, and XPA that we have discovered, and that is only revealed upon drug treatment, is significantly different from a more traditional

trigenic interaction observed in the absence of an extrinsic environmental perturbation, and merits a distinct nomenclature. However, as requested, we now describe the conceptual relevance of the trigenic screen performed in budding yeast by the Boone/Andrews labs in lines 451-454 of the revised manuscript, and cite the appropriate reference.

Reviewer #3: The authors have address my major concerns.

Reviewer #4: In the revised manuscript and accompanying rebuttal letter, the authors have adequately responded and in a satisfactory manner to all major concerns and questions of this reviewer and to the comments of the reviewers as well as fa as I can judge. With these changes and additions to the manuscript has certainly gained in quality and impact. I do have a few additional remarks, not raised before by the other reviewers, concerning mainly the DNA repair aspects.

1) On page 5, lines 143-144; the authors state that: ‘...resulting in the cancer-prone syndromes XP and CS.’, which is not correct and should be corrected. Indeed, CS is caused by a defect in a sub pathway of NER (TC-NER), but CS patients are not cancer-prone (PMID: 1308368; PMID: 19809470; PMID: 23428416; PMID: 26204423).

The reviewer is correct. We based our original statement on very old reports documenting an increased frequency of mutations in episomal plasmids that were irradiated and then passaged in CS cells. However, analysis of UV-induced mutation frequencies in CS cells using next-generation sequencing no longer supports the conclusions of those earlier papers, and as the reviewer quite correctly points out, there are no reports of an enhanced proclivity of CS patients to develop cancer of the skin or other organs. Instead, the neurodegeneration and premature aging in these patients is thought to be related to mitochondrial dysfunction, possibly related to oxidative mitochondrial DNA damage (PMID: 23435289 and 21708183).

Thank you for catching this error! We have therefore revised the text on lines 170-175 to reflect this, focusing on the utility of cell lines defective in particular NER proteins that led us to discover the MK2/XPA genetic interaction upon platinum-induced DNA damage. The revised text now reads: *“Cells genetically lacking key NER proteins involved in GG-NER have reduced ability to repair certain types of Pt-induced lesions and related types of DNA cross-links²⁸, and have persistent DNA damage and error-prone repair, resulting in the human cancer-prone syndrome Xeroderma pigmentosum (XP)³. Defects in the CSB protein, which is involved in TC-NER, base-excision repair and transcription, results in Cockayne syndrome (CS), characterized by mitochondrial dysfunction, premature aging, and neurodegeneration^{29, 30}.”*

2) lines 181-182, the authors state: “Cells depleted of either MK2 or XPA alone showed modestly increased sensitivity to cisplatin as expected...”. I do not understand this statement, as it is opposite to expectation, when XPA is depleted (which interferes with the major pathway to remove the major Cis-Pt DNA adduct (intra-strand crosslinks), as the authors state on lines 138-14300 it is expected that cells are severely sensitized towards this agent. Similarly, it is surprising to note in 2B that after depletion of XPA removal of Cis-Pt adducts is hardly affected, which again does not match with their statement that NER: ‘Cells genetically lacking key NER proteins are unable to repair Pt-induced lesions...’ (line 142). The authors should explain.

We apologize for this confusion. The assay we are using to assess sensitivity to cisplatin is a cell viability assay. Other DNA repair pathways, including translesion synthesis, homologous recombination, and the Fanconi pathway, can be used to compensate for defects in the NER pathway in repairing Pt lesions, particularly for interstrand cross-links which are thought to have a more pronounced negative effect on viability than intra-strand crosslinks. We suspect we inadvertently confused the reviewer by our overemphasis on the NER pathway in line 141.

To address this, we have therefore revised that text, which is now on lines 166-172, where we state, “Cisplatin-induced DNA damage primarily results from platinum (Pt)-mediated 1,2-intrastrand cross-links, and at a lower frequency, interstrand adducts. The former lesions are primarily repaired by the NER pathway, which includes both global genome repair (GG-NER) and transcription-coupled repair (TC-NER) (Fig 1A)²⁴, and the latter by NER, translesion synthesis (TLS), homologous recombination (HR)²⁶ and the Fanconi Anemia (FA) pathway²⁷. Cells genetically lacking key NER proteins involved in GG-NER have reduced ability to repair certain types of Pt-induced lesions and related types of DNA cross-links²⁸...”

In addition, we have removed the words “...as expected...” in the original lines 226-227, and now write, “Cells depleted of either MK2 or XPA alone showed modestly increased sensitivity to cisplatin (Fig. 2A, compare blue and red lines to black line), consistent with our prior data implicating MK2 in cell cycle arrest following DNA damage^{14, 15, 16, 17}, and the ability of other DNA repair pathways such as TLS, FA, and HR to partially compensate for defective NER.” in lines 226-229 of the revised manuscript.

3) The rebuttal to reviewer #1, point 4 is important to include in the revised manuscript, as it is indeed enigmatic to the reader (including this reviewer) why the nanoparticles would specifically target cancer cells.

Reviewer #1, point 4

The authors claim that:

MK2 has a well-established role in innate immunity and inflammatory signaling (26), similar to DNA-PK (35), raising the possibility of off-target side effects from systemic inhibition. To overcome these limitations, we developed a novel tumor-targeting nanoparticle that delivers siRNAs to the site of pre-existing MK2-containing tumors and efficiently depletes both its target RNA and protein. (page 12).

It is not easy to determine from the text why nanoparticles would target tumors specifically and not also affect immune cells and no data for this appears to be given. If the authors are using nanoparticles to avoid side effects by specifically targeting tumor cells, this should be clarified.

To experimentally address this point, and directly examine whether the nanoparticles were taken up preferentially by tumor cells, the immune cells, or both, we co-cultured 500,000 splenocytes with 500,000 KP7B tumor cells and then treated the cultures with Alexa Fluor 647 (AF647)-fluorescently tagged-siRNA-nanoplexes for 24h (Fig S3G) (c.f. Nyland, J. F., Bai, J. J. K., Katz, H. E. & Silbergeld, E. K. *In vitro* interactions between splenocytes and dansylamide dye-embedded nanoparticles detected by flow cytometry. *Nanomed-Nanotechnol* **5**, 298-304, doi:10.1016/j.nano.2009.01.001 (2009)). Viable cells were then analyzed by flow cytometry for uptake of the fluorescent siRNAs, along with identification of the specific cell populations by co-staining with fluorophore conjugated antibodies against CD45, CD11b, F480 and CD3.

CD45+CD11b+AF647+ cells were scored as siRNA+ macrophages, CD45+CD3+AF647+ cells were scored as siRNA+ T-cells and CD45-AF647+ cells were scored as siRNA+ tumor cells. This new data is now shown in Figure S3G, and described in lines 224-227 and 899-904 in the revised text. We found that the nanoplexes showed essentially no uptake in T-cells, and only very modest uptake in macrophages, compared to their uptake in the tumor cells. The molecular basis for this is not clear, but probably results from the large amount of macropinocytosis used by K-Ras-driven tumor cells to facilitate nutrient scavenging (Recouvreux and Commisso, *Front. Endocrinol.* 2017 8:261), in contrast to non-transformed cell types. *In vivo*, this tumor cell-enriched uptake is likely to be further exploited by the preferential targeting of the nanoparticles to tumors as a consequence of so-called 'passive' targeting (c.f. Harrington, K. J. *et al.* Effective targeting of solid tumors in patients with locally advanced cancers by radiolabeled pegylated liposomes. *Clin Cancer Res* 7, 243-254 (2001)). Nanoparticles are able to exploit the cancer's distinct vascular and lymphatic pathology (i.e. leaky vasculature and defective lymphatic drainage) to accumulate in the tumor. Here we utilize the enhanced tumor cell uptake and passive tumor targeting properties to formulate nanoparticles that are able to pass through the leaky vascular junctions in tumors resulting in preferential accumulation at the tumor site over time, as shown in Figure S4.

This is a good point. To address this we have now included the text that the reviewer highlighted into the body of the revised manuscript on lines 459-465.